

# The Fifth International Workshop on Ice Nucleation Phase 3 (FIN-03): Field Intercomparison of Ice Nucleation Measurements

Paul J. DeMott[1], Jessica A. Mirrielees[2,a], Sarah Suda Petters[3,b], Daniel J. Cziczo[4,c], Markus D. Petters[3,b], Heinz G. Bingemer[5], Thomas C. J. Hill[1], Karl Froyd[6,7,d], Sarvesh Garimella[4,e], A. Gannet Hallar[8,9], Ezra J.T. Levin[1,f], Ian B. McCubbin[8,9], Anne E. Perring[6,7,g], Christopher N. Rapp[c], Thea Schiebel[10,h], Jann Schrod[5], Kaitlyn J. Suski[1,i], Daniel Weber[5,j], Martin J. Wolf[4,k], Maria Zawadowicz[4,l], Jake Zenker[2,m], Ottmar Möhler[10] and Sarah D. Brooks[2]

[1]Department of Atmospheric Science, Colorado State University, Fort Collins, CO, USA
[2]Department of Atmospheric Sciences, Texas A&M University, College Station, TX, USA
[3]Department of Marine, Earth and Atmospheric Sciences, North Carolina State University, Raleigh, NC, USA
[4]Department of Earth, Atmospheric and Planetary Sciences, Massachusetts Institute of Technology, Cambridge, MA, USA
[5]Institute for Atmospheric and Environmental Sciences, Goethe University Frankfurt, 60438 Frankfurt am Main, Germany
[6]NOAA Earth System Research Laboratory, Boulder, CO, USA
[7]CIRES, University of Colorado, Boulder, CO, USA
[8]Storm Peak Laboratory, Department of Atmospheric Sciences, University of Utah, Salt Lake City, Utah, USA
[9]Department of Atmospheric Sciences, University of Utah, Salt Lake City, Utah, USA
[10]Institute of Meteorology and Climate Research (IMK-AAF), Karlruhe Institute of Technology (KIT), Eggenstein-Leopoldshafen, Germany
[a]now at: Chemistry Department, University of Michigan, Ann Arbor, MI
[b]now at: College of Engineering Center for Environmental Research and Technology (CE-CERT); Department of Chemical and Environmental Engineering, University of California Riverside, Riverside, CA
[c]now at: Department of Earth, Atmospheric, and Planetary Sciences, Purdue University, West Lafayette, IN, USA
[d]now at: Air Innova, Boulder, CO, USA
[e]now at: ACME AtronOmatic, LLC, Portland, OR, USA
[f]now at: Colorado Department of Public Health and Environment, Denver CO, USA
[g]now at: Department of Chemistry, Colgate University, Hamilton, NY, USA
[h]now at: Faculty 8- Mathematics and Physics, University of Stuttgart, Stuttgart, Germany
[i]now at: Rainmaker Technology Corporation, El Segundo, CA, USA
[j]now at: Federal Waterways Engineering and Research Institute, Karlsruhe, Germany
[k]now at: Yale Center for Law and Policy, New Haven, CT, USA
[l]now at: Brookhaven National Laboratory, Richland, WA, USA
[m]now at: Sandia National Laboratories, Albuquerque, NM, USA
Correspondence: Paul J. DeMott (Paul.Demott@colostate.edu)



**Abstract**
The third phase of the Fifth International Ice Nucleation Workshop (FIN-03) was
conducted at Storm Peak Laboratory in Steamboat Springs, Colorado in September 2015 to
facilitate the intercomparison of instruments measuring ice nucleating particles (INPs) in the
field. Instruments included a subset of two online and four offline measurement systems for
INPs, a subset of those utilized in the laboratory study that comprised the second phase of FIN
(FIN-02).  Composition of total aerosols were characterized by the Particle Ablation by Laser
Mass Spectrometry (PALMS) and Wideband Integrated Bioaerosol Sensor (WIBS) instruments,
and aerosol size distributions were measured by a Laser Aerosol Spectrometer (LAS).  The
dominant total particle compositions present during FIN-03 were composed of sulfates, organic
compounds, and nitrates, as well as particles derived from biomass burning. Mineral dust
containing particle types were ubiquitous throughout and represented 67% of supermicron
particles. Total WIBS fluorescing particle concentrations for particles with diameter > 0.5 µm
were 0.04±0.02 cm$^{-3}$ (0.1 cm$^{-3}$ highest, 0.02 cm$^{-3}$ lowest), typical for the warm season in this
region and representing ~9% of all particles in this size range as a campaign average.
The primary focus of FIN-03 was the measurement of INP concentration via immersion
freezing at temperatures > –33 °C.  Additionally, some measurements were made in the
deposition nucleation regime at these same temperatures, representing one of the first efforts to
include both mechanisms within a field campaign.  INP concentrations via immersion freezing
reported by all ice nucleation instruments generally agreed to within one order of magnitude for
measurement and sampling times coordinated to within three hours. Sometimes, much better
agreement was obtained. Outliers of up to two orders of magnitude occurred between –25 °C and
–18 °C; better agreement was seen at higher and lower temperatures. INP activity in the



immersion freezing mode was generally found to be an order of magnitude or more efficient than
in the deposition regime at 95-99% water relative humidity, although this limited data set should
be augmented in future efforts.

To contextualize the study results an assessment was made of the composition of INPs

during the late Summer to early Fall period of this study, inferred through comparison to existing
ice nucleation parameterizations and through measurement of the influence of thermal and
organic carbon digestion treatments on immersion freezing ice nucleation activity.  Consistent
with other studies in continental regions, biological INPs dominated at temperatures > –20 °C
and sometimes colder, while arable dust-like or other organic-influenced INPs were inferred to
dominate at most times below –20 °C.





**1 Introduction**


Atmospheric ice nucleation is one of the least certain aerosol-cloud interactions
influencing climate (Kanji et al., 2017). Aerosols that physically catalyze freezing, known as ice-
nucleating particles (INPs) (Vali et al., 2015), are found in the atmosphere in concentrations that
span many orders of magnitude, ranging from $10^{-3}$ L$^{-1}$ or fewer at –5 °C to 1000 L$^{-1}$ or greater at
–35 °C (Petters and Wright, 2015).  INP number concentrations typically increase exponentially
with degree of supercooling below 0 °C. However, chemical composition plays an important role
in determining if, and at what temperature, individual particles may serve as INPs (Murray et al.,
2012).  INPs initiate the formation of ice in cold and mixed-phase clouds and in turn influence
their physical and optical properties.  An increase in INP concentration over a geographic area
may increase the frequency of glaciated clouds at constant temperature, which in turn increases
precipitation and decreases cloud lifetime (Lohmann and Feichter, 2005). Nevertheless, INP
impacts on clouds simulated in global climate models are highly sensitive to how aerosol's
ability to nucleate ice is parameterized (Boucher et al., 2013). Parameterizations can only be as
accurate as the measurements on which they are based (e.g., Knopf et al., 2021).
Measurements of atmospheric INPs remain challenging due to the difficulty representing
the physical processes involved in ice nucleation instruments. At temperatures below ~ –38 °C,
micrometer-sized, dilute water droplets spontaneously freeze due to homogeneous freezing
nucleation. Homogeneous freezing nucleation is well understood and included in most cloud
formation models. However, at temperatures between 0 and –38 °C, freezing requires INPs to
facilitate nucleation through a heterogeneous nucleation mechanism (Kanji et al., 2017; Murray
et al., 2012; Vali, 1985). Nucleation is hypothesized to proceed through (1) immersion freezing,
which occurs when an INP embedded within a water droplet enters a cooler environment, and
nucleates an ice crystal, (2) condensation freezing, which occurs when freezing ensues as an



aqueous droplet condenses on the surface of an aerosol particle, (3) contact freezing, which
occurs when an aerosol in contact with a water droplet surface initiates freezing (Durant and
Shaw, 2005; Fornea et al., 2009), and (4) deposition nucleation, which is thought to occur
through the direct deposition of water vapor on an INP surface.  Of these mechanisms,
immersion freezing nucleation is thought to be the most active heterogeneous nucleation process
in the atmosphere, though there is considerable disagreement in the literature about the relative
importance of other mechanisms (Kanji et al., 2017; Ullrich et al., 2017). When the ambient
humidity is below water saturation, nucleation can occur via deposition of water from the vapor
phase. In some cases, this behavior may be ascribable instead to water condensation in pores and
cavities in aerosols facilitating freezing through a non-deposition mechanism (Marcolli, 2014;
Wagner et al., 2016). However, this process is unlikely to be of importance at temperatures > -38
°C (David et al., 2020), which are the focus of this study. We will thus refer to ice nucleation at
> –38 °C and below water saturation as happening within the "deposition regime". Study of the
efficiency of the deposition nucleation process in comparison to immersion freezing has been
limited for natural INPs.

Ice nucleation measurements have been made with instruments designed and built by

individual scientists, and more recently with commercial instruments. The ice nucleation
community has a history of collaborating to address instrument performance and inconsistencies
through participating in instrument intercomparisons, in which the custom-built instruments were
operated side-by-side, to evaluate instrument response to the same aerosol populations (Coluzza
et al., 2017; DeMott et al., 2011, DeMott et. al, 2018; Knopf et al, 2021; Lacher et al., 2024).  To
compare concentrations and compositions of INPs, a three-part workshop series, the Fifth
International Ice Nucleation Workshop, or "FIN" was held in 2014-2015. The first two phases



were held at the Karlsruhe Institute of Technology's Aerosol Interactions and Dynamics in the
Atmosphere (AIDA) facility. FIN-01 focused on determination of composition of INPs by mass
spectroscopy (Shen et al., 2024, in preparation), while FIN-02 entailed a laboratory ice nucleation
instrument comparison (DeMott et al., 2018). FIN-03, the mountaintop field intercomparison of
ice nucleation instruments is the focus of this manuscript.
Ice nucleation measurements have experienced a renaissance in the past decade, resulting
in a proliferation in both the number of custom-built instruments and a diversification of
measurement techniques employed (Zenker, 2017; DeMott, 2018; Möhler, 2021). Participation in
FIN-02 was twice that of the previous formal international workshop intercomparison in 2007 (the
International Workshop on Comparing Ice Nucleation Measuring Systems, or ICS-2007 held at
the (AIDA) facility (Jones et al., 2011; Kanji et al., 2011). During FIN-02, online and offline
instruments sampling the same population of aerosolized particles reported INP concentrations
that generally agreed within one order of magnitude across a broad temperature range. Agreement
was best in tests of immersion freezing on soils, dusts and bacteria but spanned up to 2 orders of
magnitude (or 3 °C in temperature for the same active site density) for illite NX and K-feldspar
(DeMott et al., 2018). While relatively good agreement in the laboratory between different
measurement methods during FIN-02 represents significant progress for the atmospheric ice
nucleation community, intercomparisons in ambient atmospheric settings are more difficult due to
lower typical INP concentrations (Lacher et al., 2018) and variations in the chemistry and size of
source aerosol and INPs (DeMott et al., 2017; Knopf, 2021; Lacher et al., 2024).
To evaluate how a suite of instruments operating collectively perform under the greater
measurement challenges of the field setting, FIN-03 was conducted from September 12 to 28, 2015
at Storm Peak Laboratory (SPL) in Steamboat Springs, CO, USA (Elevation: 3220 m MSL).



Unlike the subsequent closure studies of Knopf et al. (2021) and the similar comparative sampling
studies of Lacher et al. (2024), both of which occurred in regions surrounded by agricultural
activities and possible nearby urban influences, this remote continental mountaintop site at an
elevation of 3220 m provided the opportunity to sample both regional and long-range INP sources
within both the boundary layer and free troposphere. The site is typically in the free troposphere
during the nighttime and early morning, and in the boundary layer from the late morning to early
evening, although topography and wind direction influence this timing (Collaud Coen et al., 2018).
When in the free troposphere, the site is more likely to reflect influences by regional or long-range
transport of aerosols. For example, during FIN-03, the variety of air masses that were sampled and
sensed by aerosol instruments included ones passing over phosphate mines in Idaho (on September
18 and 20) and mined deposits of rare earth metals at Mountain Pass, CA (on September 27)
(Zawadowicz et al., 2017). When the convective boundary layer height reaches the elevation of
the laboratory, the site is likely more impacted by local/regional aerosol sources. Additionally,
meteorological transitions can occur (e.g., frontal boundary passage, wind direction shifts), driving
changes in aerosol sources that may indirectly occur in response to those changes (e.g., biological
aerosols, carbonaceous particles from biomass burning, and mineral/soil dust). While the
constantly fluctuating environmental conditions during FIN-03 added an additional challenge to
the intercomparison, they also provided a realistic setting for atmospheric INP measurements. In
addition to adding challenges, conducting the intercomparison in the presence of complex aerosols
in the field provided the opportunity to survey instrument response to varied aerosol sources.

Participation in FIN-03 included online continuous flow diffusion chambers (CFDCs) and

aerosol collections for offline INP measurements, representing a subset of the instruments that
operated in FIN-02 (DeMott et al., 2018). Since aerosol physical and chemical properties strongly





influence their ability to activate as INPs (Hoose and Möhler, 2012; Kanji et al., 2017; Murray et
al., 2012), measurements of aerosol sizes and composition (see Section 2) were included to lend
context to the variable composition of aerosols and evaluate their potential role in ice nucleation
activity. Rather than use these data for attempting closure, FIN-03 focused on using data to
constrain existing parameterizations to diagnose INP compositions during the study period. Also,
in contrast to other recent studies, special effort was made to characterize deposition nucleation
activity in addition to immersion freezing.
**2 Methods**
**2.1 Aerosol property measurements**

Measurements of aerosol physical, chemical, and biological particle properties were

made during FIN-03 to provide context to INP measurements. Sampling manifolds, which draw
air into SPL from outdoors at high flow, are as follows: Inlets were located in each of the two
wings of SPL that frame the living area, referred to as the "instrument" laboratory (facing north)
and the "chemistry" laboratory (facing south). The "original" inlet system in the instrument
laboratory (Hallar et al. 2011; Petersen et al. 2019) feeds a nephelometer (see below) and a
standard suite of aerosol instruments (not operational for FIN-03). This 15 cm diameter
aluminum inlet rises 4 m above the roofline. At ~1 m inside the laboratory, it transitions to a 15
cm horizontal manifold. With a flow of ~500 L min$^{-1}$, aerosol transmission calculations have
characterized the system to have a 50% upper particle size cut-off at an aerodynamic diameter of
5 µm (Hallar et al., 2011). The "new" inlet system consists of two identical stainless steel,
turbulent-flow, ground-based inlets described by Petersen et al. (2019), which are straight and
enter the laboratory vertically. One is in the SPL instrument laboratory, and one is in the
chemistry laboratory. These inlets that extend 10 m above the laboratory roof have been





demonstrated to have 50% upper particle size cut-offs at an aerodynamic diameter of
approximately 13μm for a wind speed of 0.5 m s$^{-1}$. Additional computational fluid dynamics
simulations suggest that this size cut off remains above 5 μm even for exterior wind speeds up to
15 m s$^{-1}$ (Petersen et al., 2019), higher than achieved at any time during FIN-03 sampling. Little
bias was seen in ambient aerosol sampling between the original inlet system and the new,
turbulent flow-based inlets based on the metric of total aerosol scattering (Petersen et al., 2019).
Flow rates and transfer lines to individual instruments are described after the aerosol property
measurements are introduced, at the conclusion of this section.

A Laser Aerosol Spectrometer (LAS, model 3340, TSI Inc., St. Paul, Minnesota, USA)

was used to measure the aerosol size distribution over the diameter range 0.089-10 μm. Aerosols
were assumed dry based on relative humidity always remaining below 30% when measured from
its sample line. Sample was drawn at 0.1 L min$^{-1}$ and sampling was done from the turbulent flow
inlet system located in the SPL chemistry laboratory, as described further below. Size
calibrations were performed using polystyrene latex spheres (PSL, Duke Scientific).  PSL
diameters were converted to ammonium sulfate equivalent diameters using Mie theory (Froyd et
al., 2019). Particle concentrations are reported as a function of equivalent ammonium sulfate
diameter. Volume and surface area distributions are derived assuming spherical particles.
Number concentrations and surface areas, further informed by aerosol composition
measurements, allows for connection to INP concentration predictions, and this information is
used herein to diagnostically infer mineral and soil dust influences on INPs during the study. We
will particularly reference the parameterizations of Niemand et al. (2012) that links mineral
surface area to INP concentrations and DeMott et al. (2015) that links dust number
concentrations at sizes larger than 0.5 μm to INP concentrations.



Measurements using a three-wavelength integrating nephelometer (TSI Model-3563,
Shoreview, MN) also provided information on aerosol distributions via their optical properties.
This nephelometer is part of the National Oceanic and Atmospheric Administration Federated
Aerosol Network (Andrews et al., 2019). The nephelometer splits scattered light into red (700
nm), green (550 nm), and blue (450 nm) wavelengths. Impactors to cut aerosols at aerodynamic
sizes below 1 and 10 µm are alternately used upstream of air flowing into the instrument. The
nephelometer sampled within the original inlet in the SPL instrument laboratory. A blunt tap
from this original SPL inlet manifold provided air samples to the nephelometer system via 1" i.d.
conductive tubing.
The Particle Analysis by Laser Mass Spectrometry (PALMS) instrument performed
measurements of the composition of 0.2 to 3.0 µm aerosol particles. The PALMS was designed
and operated by the National Oceanic and Atmospheric Administration (NOAA) as described in
Thomson et al. (2000). Particles are sampled, focused, and accelerated via an aerodynamic lens
inlet (Schreiner et al., 2002) before passing into a vacuum chamber where they successively pass
through two continuous-wave detection laser beams (532 nm Nd:YAG) and scatter light.
Vacuum aerodynamic diameter is determined based on the transit time. The detection signal
triggers an ArF excimer laser that emits a 193 nm pulse to simultaneously ablate and ionize
single particles. The resulting ions are analyzed with a unipolar time-of-flight mass spectrometer,
which allows polarity switching during the particle flight and thereby producing positive or
negative mass spectra for individual particles. PALMS spectra are classified into compositional
categories, and fractions are averaged over 5 min sample periods. Number, surface area, and
mass concentration products for the different particle types are generated by combining PALMS
size-dependent fractional composition data with absolute particle concentrations measured by the




LAS instrument (Froyd, et al. 2019; Froyd et al., 2022). When PALMS compositional
concentrations are referenced in the results of FIN-03 aerosol compositions in Section 3.2, they
have been determined by these methods.
The NOAA Wideband Integrated Bioaerosol Sensor, Model 4A (WIBS-4A; Droplet
Measurement Technologies, Longmont, CO) was used to detect fluorescent properties of
individual particles and assess the presence of biological particles.  Measurements are presumed
to characterize dry particles. The WIBS-4A is described in detail elsewhere (Gabey et al., 2010;
Kaye et al., 2005; Perring et al., 2015) and is only briefly summarized here. As described in
Zawadowicz et al. (2019), the gain for the WIBS-4A used at SPL was set to detect and classify
particles between 0.4 and 10 μm.  First, the optical diameter of particles entering the detection
cavity is determined by light scattered during transit through a 635 nm laser beam. This signal
triggers the sequential firing of two xenon flash lamps filtered to produce narrow excitation
wavebands centered at 280 and 370 nm. The resulting fluorescence is detected by two wideband
photomultiplier detectors observing 310-400 nm and 420-650 nm. Fluorescing particles were
categorized according to the intensity of the signal in each of three channels (channel A
excitation 280 nm/emission 310-400 nm, channel B excitation 280 nm/emission 420-650 nm,
channel C excitation 370 nm/emission 420-650 nm). Particles for which the measured emission
intensity in only one channel met the threshold (such that the signal intensity exceeded the value
equal to three standard deviations above the mean) were assigned Type A, B, or C, and particles
for which the measured emission intensity in two or more channels met the threshold were
assigned Type AB, BC, BC, or ABC (Perring et al., 2015).  The interpretation of particle
composition according to the seven WIBS-4A channels is not straightforward, as many
fluorophores are active in each channel, including non-biological components (Perring et al.,





2015; Pöhlker et al., 2012). Channel A fluorophores include biological components such as
tryptophan, phenylalanine as well as nonbiological components which interfere with the
determination of biological content, including polycyclic aromatic hydrocarbons (PAHs)
(pyrene, naphthalene, phenanthrene). Biological fluorophores, which produce a signal in channel
C, include the reduced form of nicotinamide adenine dinucleotide (NADH), nicotinamide
adenine dinucleotide phosphate (NADPH), and riboflavin, and potential non-biological
interference in channel C may result from the presence of humic acid in aerosol particles.
Channel B fluorophores are not generally considered to be biological in nature, though riboflavin
and dry cellulose both produce signals in this channel.
We report WIBS-4A channel data herein under these noted caveats and further utilize
these data to explore links to immersion freezing biological INP concentrations, as has been done
in some previous efforts. Tobo (2013) previously reported relations of biological INPs acting in
the immersion freezing mode (measured by the CSU CFDC) to fluorescent biological aerosol
particles (FBAP) at sizes > 0.5 μm measured in the understory of a Ponderosa pine forest in
Colorado. In that work, an ultraviolet aerodynamic particle sizer (UV-APS) with excitation
wavelength at 355 nm and emission wavelengths 420-575 nm was used as a reference for FBAP
concentrations. Due to differences between the excitation and emission wavelengths, UV-APS
measurements correspond most closely with Type C particles detected by the WIBS-4A (Healy
et al., 2014). Consequently, a conservative or "low" estimate of FBAP for use in the
parameterization of Tobo et al. (2013) we employ herein uses the sum of C, AC, BC and ABC
particles. A "high" FBAP for this parameterization has also been used by Twohy et al. (2016),
considering all non-B-only particles (A, AB, ABC, AC, BC, C). We will use both definitions in
our presented results and partly justify the higher estimate because the CSU CFDC assuredly





does not capture all biological INPs due to the use of the upstream impactor. A final class of
particles defined by WIBS-4A data for relation to immersion freezing INPs are denoted as FP3
particles (Wright et al., 2014). FP3 particles are particles that show strong emission in the 310 to
400 nm spectral band when excited by 280 nm light (A type) but are only weakly represented as
B and C types. A threshold of 1900 arbitrary fluorescence units in the 310 to 400 nm band is
used to denote FP3 particles (Wright et al., 2014). FP3 particles have been connected to
immersion freezing INP concentrations in multiple environments (Wright et al., 2014; Suski et
al., 2018; Cornwell et al., 2023).

Flow rates and transfer lines to each instrument are summarized as follows. The PALMS,

LAS, and WIBS-4A sampled from the SPL turbulent flow inlet stack at 0.75, 0.1, and 0.3 vlpm,
respectively ,via a common 1/4" o.d. aluminum tube. The total flow was held at 1.2 vlpm using
a variable dump flow, and the line was split into multiple 1/8" o.d. stainless steel tubing sections
connecting to each instrument. All tubing junctions employed Y-splitters, and all reducing
fittings were internally beveled to prevent impaction losses. Sample lines were not actively dried,
but relative humidity was < 30% in LAS and WIBS-4A. For the LAS instrument, the theoretical
transmission of the inlet system was 98%, 84%, and 57% for 1, 3, and 5 μm aerodynamic
diameter particles, respectively, with gravitational settling being the dominant loss process.
Transmission to WIBS-4A for the same sizes was 99%, 90%, and 76%. Size distributions were
not corrected for transmission losses. The nephelometer sampled from the original inlet in the
SPL instrument laboratory via a blunt tap manifold and 1" i.d. conductive tubing.






**Table 1** Descriptions of INP instruments.

|  | Instrument | Type | Institute | References |
|---|---|---|---|---|
| Online/direct | Continuous flow diffusion chamber (CSU-CFDC) | Continuous flow diffusion chamber (cylindrical) | Colorado State University | (Eidhammer et al., 2010; Rogers, 1988; Rogers et al., 2001) |
|  | Spectrometer for ice nuclei (MIT-SPIN) | Continuous flow diffusion chamber (parallel) | Massachusetts Institute of Technology | (Garimella et al., 2016; Garimella et al., 2017; Kulkarni & Kok, 2012) |
| Offline/post-processing | Frankfurt Ice Nuclei Deposition Freezing Experiment deposition mode (FRIDGE-DEP) | Low pressure diffusion chamber (on wafers) | Goethe University Frankfurt | (Schrod et al., 2016) |
|  | Frankfurt Ice Nuclei Deposition Freezing Experiment immersion freezing mode (FRIDGE-IMM) | Cold stage droplet freezing array (on wafers) | Goethe University Frankfurt | (Schrod et al. 2020; DeMott et al. 2018) |
|  | Ice spectrometer (CSU-IS) | Aliquot freezing array | Colorado State University | (Hill et al., 2016; Hiranuma et al., 2015) |
|  | Cold stage (NC State-CS) | Cold stage droplet freezing array (on hydrophobic glass slides) | North Carolina State University | (Wright & Petters, 2013; Yadav et al., 2019) |


**2.2 INP measurement methods**

A combination of direct-processing (online) and post-processing (offline) ice nucleation

instruments were employed during the FIN-03 field campaign. All these instruments were also
used in the FIN-02 laboratory campaign. Online instruments have the advantage in that the
aerosol being evaluated as INPs remain free-floating and unaltered, never touching a substrate
nor requiring shipment of samples to a laboratory. Online techniques can also monitor INP



concentration changes occurring over short time scales. Nevertheless, they are limited in the
thermodynamic conditions that can be represented over a given time frame, and they are limited
by volume sampling rates in assessing the low concentrations of INPs at modest supercooling.
Offline techniques, i.e., those in which samples are collected in the field and subsequently
processed in laboratory, provide the opportunity to capture large sample volumes (albeit over
longer time scales) and consequently assess a wider temperature range of INP activation
properties. A summary listing of all ice nucleation instruments is provided in Table 1. Detailed
operating principles, siting of samplers (rooftop versus within SPL), and experimental methods
for each instrument follow below. In this work, we will refer to the FIN-03 "intercomparison
period" to define the times that all INP instruments co-sampled air with substantial temporal
overlap for direct comparison. Other times of sampling by the different instrument groups were
devoted to special science investigations that are not covered herein.

*2.2.1 Online INP measurements*

Two online instruments participated in intercomparison experiments in FIN-03. One, the

Colorado State University CFDC (CSU-CFDC), has the most established history as an online
technique for activating and counting INPs.  The CSU-CFDC operating principles are described
in several prior works (Rogers, 1988; Rogers et al., 2001; Eidhammer et al., 2010). Application
and considerations for interpreting data have been described in detail in several publications,
most recently by DeMott et al. (2018). The CSU-CFDC is composed of nested cylindrical copper
walls that are chemically ebonized to be hydrophilic so they can be evenly coated with ice. The
chamber is divided into two sections vertically. For FIN-03, the CSU-CFDC was operated to
establish a temperature gradient between the colder (inner) and warmer (outer) ice walls in the



upper ~50 cm "growth" section to produce either water subsaturated or water supersaturated
conditions at various temperatures within a central lamina. For the flow rates used (10 vlpm total
flow, 1.5 vlpm sample flow) the residence time was ~5 s in the growth region. Aerosol particles
were directed into that central lamina. Ice crystals forming on INPs in the growth section
continued to grow for ~2 s in the lower ~35 cm "evaporation" section of the chamber where the
outer wall temperature was adjusted to be at an equivalent temperature to the inner (cold) wall to
promote evaporation of liquid drops. When operating in the water supersaturated regime, water
relative humidity was controlled to be nominally at 105% during FIN-03 to stimulate droplet
growth and subsequent freezing, for best comparison to offline immersion freezing methods. For
probing ice nucleation in the deposition nucleation regime, relative humidity (RH) was
controlled to ~95%.

The CSU-CFDC sampled from one of the turbulent aerosol inlet ports, located in the SPL

instrument laboratory. Connection was via ¼" o.d. conductive tubing. Prior to entering the CFDC,
aerosol was further dried using two inline diffusion driers and then size-limited using dual single-
jet impactors that achieve a 50% upper particle size cut-off at an aerodynamic diameter of 2.5 μm.
This limitation on aerosol sizes helps to remove ambiguity when distinguishing ice crystals at ~4
μm sizes from aerosol particles using an optical particle counter at the CSU-CFDC outlet.
Temperature uncertainty is ± 0.5 °C at the reported CSU-CFDC lamina processing temperature
and relative humidity uncertainty depends inversely on temperature, as discussed by DeMott et al.,
(2018), estimated for example as 2.4 % at –25 °C.

To correct for background counts that can occur due to ejection of frost emanating from

interior surfaces of the CSU-CFDC over operational periods, and for defining measurement
uncertainties, we follow Levin et al. (2019). Frost corrections are defined through intervals of





sampling ambient air through a HEPA filter. Sample data were background corrected by
subtracting the interpolated filter period concentration before and after each sampling period.
Background corrected data were then averaged to ~5-min sampling times to increase statistical
confidence. Poisson counting errors during filtered and ambient sampling periods were added in
quadrature, and INP concentrations were judged statistically significant at the 95% confidence
level if they were greater than 1.64 times this combined INP error (one-tailed z test). Interior inlet
tubing losses are not considered in the reported INP data because they have been estimated at 10%
or less in the past. INP concentration correction underestimates inferred (by 3 time) to be due to
aerosols spreading outside of the lamina during measurements specifically of mineral dust INPs
(DeMott et al., 2015) are not generally applied to the data herein, though this is discussed regarding
INP parameterizations in this paper.

An aerosol concentrator (MSP Model 4240) was used at selected times during FIN-03 to

improve sampling statistics, in the same manner as in previously published studies (Tobo et al.
2013; Suski et al., 2018; Cornwell et al., 2019). The aerosol concentrator was positioned open to
the air on the roof of the instrument laboratory room (covered and not used during rainfall), with
a short ¼" o.d. copper line containing the concentrated aerosol entering the laboratory from about
3 m above the CFDC. Concentration factors were evaluated in the same manner as Tobo et al.
(2013), leading to an average increase of INPs by 90 times during operation of the aerosol
concentrator compared to ambient inlet periods during the study (not shown here because analysis
repeats numerous past efforts).

A three-way manual stainless-steel valve was used to direct sample air to the CFDC from

either the turbulent flow inlet or the aerosol concentrator. At times, a high temperature heating
tube (Suski et al., 2018) or a single particle soot photometer (Schill et al., 2016) was placed in-line



following the three-way valve for removing aerosol organics or black carbon (not reported here)
prior to INP measurements. The use of a tube heater upstream of the CSU CFDC to expose single
particles to 300°C, following the methods of Suski et al. (2018), is intended to isolate the action
of total organic versus inorganic INPs via comparison of ambient versus heat-treated particle
streams. Simultaneous measurements of both aerosol streams is of course not possible with a single
CFDC, so sampling was conducted by alternating the inlet chosen during subsequent 10-minute
periods, and ignoring any very short-term aerosol changes that might occur over such times. This
was a special contribution by the CSU CFDC group, for comparison to bulk aerosol treatments
discussed in the next subsection.

A second online instrument, the SPectrometer for Ice Nuclei operated at the time by the

Massachusetts Institute of Technology (MIT-SPIN; Droplet Measurement Technologies, Boulder,
CO), a commercially produced, parallel-plate continuous flow diffusion chamber style instrument,
also sampled during FIN-03. Operating principles are described in Garimella et al. (2016) and
Garimella et al. (2017). SPIN consists of two flat walls separated by 1.0 cm and coated in
approximately 1.0 mm of ice. Aerosol particles are fed into the chamber in a lamina flow of about
1.0 liters per minute and are constrained to the centerline with a sheath flow of about 9.0 liters per
minute. The temperature and relative humidity that the aerosol lamina experiences are controlled
by varying the temperature gradient between the two iced walls (Kulkarni & Kok, 2012). After
exiting the nucleation chamber, the particles enter SPIN's optical particle counter, which sizes
aerosol on a particle-by-particle basis for diameters between 0.2 and 15 μm. For FIN-03, the SPIN
sampled from one of the turbulent flow inlet systems, located within the SPL aerosol chemistry
laboratory. It was connected to the inlet system port with a short section of ¼" o.d. conductive
tubing.



Data processing for SPIN was performed following a similar procedure as the CSU-CFDC
instrument. Particle counts from the OPC were first filtered to only consider particles larger than
5 μm. A low-pass filter was applied next to remove all 1 Hz data that exceeded a total of three
counts. Particle data was then converted from counts per second to number density (n $L^{-1}$) using
the combined sheath and sample flow exiting the OPC. A SPIN specific particle concentration
correction factor of 1.4 is applied to account for non-ideal instrument behavior resulting in
underestimation of INP as described by Garimella et al. (2017). As the field measurements from
this study predate the laboratory experiments performed to determine SPIN uncertainties, we select
the minimum reported correction factor to remain conservative in our measurements. A
depolarization filter was then applied to isolate particle data specific to ice using 1 Hz averaged
backscattering data from the SPIN's OPC, with instrument specific values of 3.5 and -0.25 for the
$\log_{10}(Size)$ and $\log_{10}(S1/P1)$ measurements respectively. Frost ejected from the plates of the SPIN
chamber was characterized by particle-free sampling periods when the sample flow was diverted
through a HEPA filter by an automated three-way valve. Linear interpolation was used to
approximate background frost throughout the measurement period and smoothed using a five-
minute moving average. Sample data was background frost corrected by subtracting this smoothed
background frost density from total number density. Lastly, data points that exceeded water
saturation were excluded from analysis.
Estimation of measurement error for the MIT-SPIN follows a similar procedure to the
CSU-CFDC. Assuming the background corrected INP concentration follows a Poisson
distribution, the Poisson error for both INP and background frost concentrations were defined as
the square root of the sample mean. The significance test statistic was defined by the quadrature
sum of counting errors multiplied by the z-score for a one-tailed z-test at the 95% confidence



interval. INP measurements were deemed statistically significant if the mean INP concentration
was greater than this test statistic.

A third online instrument, the Texas A&M CFDC that shares the same design aspects of

the CSU CFDC, was used for special studies conducted outside of intercomparison exercises
(Zenker et al., 2017).
***2.2.2 Offline INP measurements***

Offline methods have undergone many improvements in recent years and have been

successfully demonstrated for being used in a complementary manner to online methods in other
recent intercomparisons (DeMott et al., 2017; DeMott et al., 2018; Hiranuma et al., 2015; Wex et
al., 2015). In FIN-03 samples were collected with liquid impingers and filter samplers and
analyzed for immersion freezing of distributed liquid particle suspensions using the North
Carolina State University Cold Stage (Wright et al., 2013), the CSU Ice Spectrometer (Hiranuma
et al., 2015; DeMott et al., 2018), and the FRankfurt Ice Nuclei Deposition FreezinG Experiment
(FRIDGE) instrument (Schrod et al., 2016).

The North Carolina State University Cold Stage (NC State-CS)

The North Carolina State University cold stage (NC State-CS), previously described by

Wright and Petters (2013) and Hader et al. (2014). Procedures used for collecting immersion
freezing spectra are described below and by Yadav et al. (2019). During FIN-03, filter samples,
impinger samples and precipitation samples were collected for analysis using the NC State-CS.
For the intercomparison, the filter and impinger results are considered. Filter samples were
collected from the roof of Storm Peak Lab for 3–4 hrs twice daily using 47 mm Nuclepore
polycarbonate filters (0.2 µm pore size) housed in an open-faced stainless-steel filter holder
operated at 14 L min$^{-1}$ (at altitude) or ~9 L min$^{-1}$ at standard temperature and pressure conditions



(STP) of 1013 mb and 0 °C. Filter holders were directed downward and sheltered from
precipitation by a large, inverted metal bowl. Images are shown in supplemental Section S1.
Impinger samples were collected directly into water using a glass bioaerosol impinger (SKC, Inc.)
as described by Hader et al. (2014) and DeMott et al. (2018). The impinger jets air at 10.6 L min⁻
$^1$ (~7 L min$^{-1}$ STP) into a 20 mL water reservoir, impacting 80% of particles ≥ 200 nm in diameter
and ~100% of particles ≥ 1 µm (Willeke et al., 1998). Impinger samples were collected in the same
manner as was done for all shared liquid samples for the FIN-02 intercomparison (DeMott et al.,
2018) excepting that Teflon tape replaced stopcock grease to seal the impinger glass lid to prevent
jamming. Water evaporating from the reservoir was replaced hourly; the impinger was in a rooftop
shelter with its inlet extending through a hole in the shelter wall, into the open air at a height of ~6
feet below the position of filter sampling units that were mounted on an outside railing. Water used
onsite was filtered (0.2 µm) Milli-Q water. All samples were stored at –20 °C onsite, shipped on
dry ice, then stored at –80 °C until analysis.

Freezing statistics for each liquid sample were acquired by pipetting an array of

approximately 256 droplets of 1 µL ± 0.88% volume on four hydrophobic glass slides under dry
N$_2$ gas. Temperature was ramped at a rate of –2 °C min$^{-1}$ and freezing was detected optically by a
microscope at a temperature resolution of 0.17 °C (every 5 s). Freezing temperature spectra are
expected to be independent of cooling rate (Wright et al., 2013). The concentration of ice nuclei
at temperature $T$ per unit volume of liquid is given by Vali (1971):

$$c_{IN}(T) = \frac{-ln\left(f_{unfrozen}(T)\right)}{V_{drop}\Delta T} \tag{1}$$

where $f_{unfrozen}$ is the fraction of unfrozen droplets at $T$ and $V_{drop}$ is the population-median droplet
volume. The concentration of ice nucleating particles (INP) in the atmosphere is given by:



$$c_{INP}(T) = \frac{c_{IN}(T) \cdot f \cdot V_{liquid}}{V_{air}} \tag{2}$$

where $f$ accounts for sample dilution, $V_{liquid}$ is sample volume, and $V_{air}$ is the volume of air

sampled (flow rate at STP × duration). Freezing spectra were collected 3× per sample and binned

into 1 °C intervals. Confidence intervals reported in archived data are ±2 standard deviations of

the mean. We will refer to the processed filter samples as NC State-CS (F) and processed

impinger samples as NC State-CS (I).

#### CSU Ice Spectrometer (CSU-IS)

The CSU-IS also post-processed particles sampled onto filters during FIN-03. This

instrument has been described in Hiranuma et al. (2015) and Suski et al. (2018). Samples were

collected for approximate periods of 4 hours for intercomparison periods (longer for overnight

samples – not part of the intercomparison) using pre-cleaned 0.2 μm pore diameter, 47 mm

polycarbonate Nuclepore filters (Suski et al., 2018) mounted in disposable, sterile open-faced and

face-up holders (Nalgene), with a typical sample flow rate of 14.9 L min$^{-1}$ (ambient) and 9.5 L

min$^{-1}$ (STP). Filters were collected on the same exterior laboratory roof railing as the NC State

filters, approximately 2 m distance away. All filter samples were frozen following collection, and

until processing at CSU. Pre-sterilization procedures and overall clean protocols for preparation

and handling of filters are detailed in Suski et al. (2018). Particle re-suspension was done through

20 minutes of shaking filters in sterile 50 mL Falcon polypropylene tubes (Corning Life Sciences)

with 6-10 mL of 0.02 μm pore diameter filtered deionized water. Further 20-fold dilutions using

filtered water were made as needed to permit measurement of freezing spectra to the low

temperature limit of operation of the CSU-IS.

Immersion freezing INP temperature spectra were obtained by distributing 24 - 32 aliquots

of 50 μL particle suspensions into the sterile 96-well PCR trays that mount in the CSU-IS. The





cooling rate was –0.33 °C min$^{-1}$. Frozen wells were counted at 0.2 - 1 °C degree intervals to a limit
of about -28 °C, and cumulative numbers of INP mL$^{-1}$ of suspension estimated following Vali
(1971) and Eq. 1. Conversion to ambient air concentrations std L$^{-1}$ were made based on distributed
suspension volume and the total air volumes collected (Eq. 2). Filter blanks were collected during
FIN-03, one was tested and used to obtain background INP numbers per filter. Blank INPs were
found to account for <5% of INPs at –20 and –25 °C. Binomial sampling confidence intervals
(95%) were derived for INP concentrations following Agresti & Coull (1998). The temperature
uncertainty of INP measurements is ±0.2 °C.

As a special contribution to FIN-03, portions of IS aerosol suspensions were set aside (e.g.,

suspensions of 6 to 10 ml can serve up to three or more IS aliquot fills) for treatments to proximally
isolate total biological, other organic and inorganic contributions to measured immersion freezing
INP concentrations. To assess removal of heat labile INP entities, a 2 mL aliquot of suspension
was re-tested in the IS after heating to 95 °C for 20 min (McCluskey et al. 2018). To attempt to
remove all organic INPs, 1 mL of 30% $H_2O_2$ was added to a 2 mL aliquot of suspension and the
mixture heated to 95 °C for 20 min while illuminated with UVB fluorescent bulbs to generate
hydroxyl radicals (residual $H_2O_2$ is then removed using catalase) (Suski et al. 2018), and the INPs
were again assessed for freezing spectra in the IS. Herein we describe a subset of samples collected
on September 15, September 23, and September 25 that were subjected as IS suspensions to the
two treatments.  These treatments are based on well-established methods which have been used to
assess biological components in samples for more than 60 years (Alsante et al., 2023, and
references therein).  The interpretation of data from exposure of particle suspensions to 95 °C is
that the reduction of INP concentrations under thermal treatment is a proxy for the concentration
of biological (proteinaceous and microbial) INPs which have been eliminated or deactivated





through treatment. A strong reduction in INP activity observed after peroxide treatment indicates
dominant organic INP populations, whereas a lack of response to this treatment is assumed to
indicate that inorganic INPs such as mineral dusts dominate non-heat labile INPs. This assessment
for bulk suspensions of particles could be directly compared to measurements of 300C heat treated
single particles in the online CSU CFDC measurements on these same days, providing a more
insightful investigation of INP compositions.

The use of such treatments and the insights they convey for atmospheric ice nucleation

studies has been reported in published studies of INPs for a variety of locations (McCluskey et al.,
2018; Suski et al., 2018; Barry et al., 2021a; Knopf et al., 2021; Testa et al., 2021). Taken together,
such treatment studies show general utility for estimating biological contributions to INP, overall
organic contributions and the importance of inorganic contributions. However, we note that not all
biological materials may be completely denatured or removed by heat (Testa et al., 2021; Daily et
al., 2022; Alsante et al., 2023) and not all organics may be removed by peroxide.  For example,
denaturation is the disruption of higher order (secondary, tertiary, and quaternary structure) in a
protein which leads to a loss or lessening of function. It follows that simpler proteins or peptides,
such as glutathione, have no higher order structure, and thus cannot be denatured (Alsante et al.,
2023).  Consequently, estimates of biological contributions to INP based on these treatments may
be considered as lower limits.

FRIDGE Cold Stage and Deposition Nucleation Measurements

The FRIDGE instrument can be used to measure the concentration of INPs by two

independent methods: a) a droplet freezing assay on a cold stage (hereafter: FRIDGE-CS Schrod
et al., 2020; DeMott et al. 2018; Hiranuma et al. 2015) which addresses immersion freezing
similarly to the NC State-CS and the CSU-IS and b) the diffusion chamber method (hereafter:



FRIDGE-DC), that addresses the deposition nucleation and condensation freezing modes
introduced in Schrod et al. (2016) and referred to as the "standard" method in previous publications
(e.g., DeMott et al, 2018). The ice nucleation analysis is performed inside the FRIDGE instrument
for both methods, yet the sampling process, addressed nucleation modes and the specific analytical
procedure differs as described below.
For the FRIDGE-CS method, aerosol particles were sampled via a short ¼" conductive
tube from the shared turbulent flow aerosol inlet in the SPL instrument laboratory on Teflon
membrane filters (Fluoropore PTFE, 47 mm, 0.2 μm, Merck Millipore Ltd.). The sampling
duration ranged from 50 to 240 minutes, resulting in air volumes between 250 and 1000 std. L.
The particles were extracted in 10 ml deionized water by shaking. Approximately 150, 0.5 μL
droplets from that solution were pipetted onto a clean, silanized silicon wafer on the cold stage of
the FRIDGE instrument and cooled by $-1°C$ $min^{-1}$ at ambient pressure. A CCD camera detects
freezing events and counts the number of frozen droplets as a function of temperature. This process
is repeated with fresh droplets and fresh substrates until approx. 1000 droplets are attained. The
INP number concentration is derived by Eqs. 1 and 2.
For the FRIDGE-DC measurements, an electrostatic aerosol collector EAC (Schrod et al.,
2016) was connected to the same aerosol flow inlet via a short ¼" conductive tube. The EAC
consists of a cylindrical sampler, whose inlet is concentrically surrounded by 12 gold wires that
are at 12 kV against a grounded silicon wafer, which is used as the sample substrate, at the
bottom of the sampler. Once the airflow is pumped inside, aerosol particles are charged by
electrons emitted from the gold wires and are precipitated onto the silicon wafer. The analysis at
certain pairs of T and RH follows in a separate step. For that, the wafer is placed on the cold
stage inside a diffusion chamber. After the chamber is evacuated, the temperature is set to the



first analysis temperature. In a second, much larger volume, pure water vapor is regulated by
pressure control to the desired supersaturation. Once the water vapor diffuses into the chamber,
ice forms on the activated INPs and a CCD camera is used to record and count the emerging ice
crystals, which appear as bright objects. It is assumed that one ice crystal represents one INP.
The water vapor atmosphere and thus the growth of ice crystals is maintained for up to 100
seconds until the valve to the water vapor source is closed and the chamber is evacuated again.
The process is repeated at increasing humidity first, and then at progressively lower
temperatures. At SPL samples were taken from the electrostatic sampler for 50, 75 and 120
minutes, resulting in volumes of approximately 64-150 sL. The samples were analyzed by
default at –20 °C, –25 °C and –30°C and 95 %, 99% and 102% water saturation. In addition, a
few samples were analyzed at –15 °C. This was a special contribution by the FRIDGE group for
comprehensive analysis of INP activation in the deposition regime, and for comparison to online
data in this regime collected for some days.
**2.3 INP processing and sampling strategies**

As a campaign strategy, samples were collected over different time periods in the day to

reflect both varied weather conditions and aerosol populations arriving at the mountain
laboratory. For intercomparison, a select number of 4-hour sampling periods were allocated in
which online instruments attempted to operate at a few predesignated temperature and relative
humidity ranges, while samples were collected continuously for off-line analysis. While aerosol
conditions can change within a 4-hour time frame, this was agreed upon as a minimal reasonable
period for comparability. Similar sampling strategies have been employed in the past
intercomparisons (DeMott et al., 2017; Knopf et al., 2021). Overall, measurements were
conducted over a wide range of temperatures (–7 to –34 °C) in the heterogeneous ice nucleation



regime.  Sampling such a broad data set allows for the consideration of instrument performance
in response to the presence of highly active INPs (i.e., those facilitating freezing at temperatures
≥ –10 °C) as well as more modestly effective INP (–10 °C ≥ T ≥ –30 °C). In addition, the range
of INP concentrations includes lower concentrations which challenge instrumental limits of
detection.
**3 Results and discussion**
**3.1 Meteorological context**

Weather conditions during FIN-03 were characterized using auxiliary measurements.

Weather data (temperature, humidity, winds and pressure) were obtained for Storm Peak
Laboratory through the MesoWest ([https://mesowest.utah.edu/cgi-](https://mesowest.utah.edu/cgi-)
[bin/droman/meso_base_dyn.cgi?stn=STORM](https://mesowest.utah.edu/cgi-bin/droman/meso_base_dyn.cgi?stn=STORM)) mesonet (STORM site), supplemented with
measurements from instruments operated at SPL through the Western Regional Climate Center
(WRCC) ([https://wrcc.dri.edu/weather/strm.html](https://wrcc.dri.edu/weather/strm.html)) for the two days that were absent in the
MesoWest record. Air temperature, relative humidity, and barometric pressure time series are
shown in Figure 1(a), 1(b) and 1(c), respectively. Precipitation was measured via a rain gauge at
Storm Peak Laboratory provided by NC State. Precipitation rate was calculated from the quotient
of precipitation (in mm) and time collected (in hours), as shown in Figure 1(d). Back trajectories
for all the sampling days in FIN-03 are reported by Zawadowicz et al. (2017), showing 72-hr air
mass transits from regions that included Southern California, Washington State and Eastern
Nebraska.

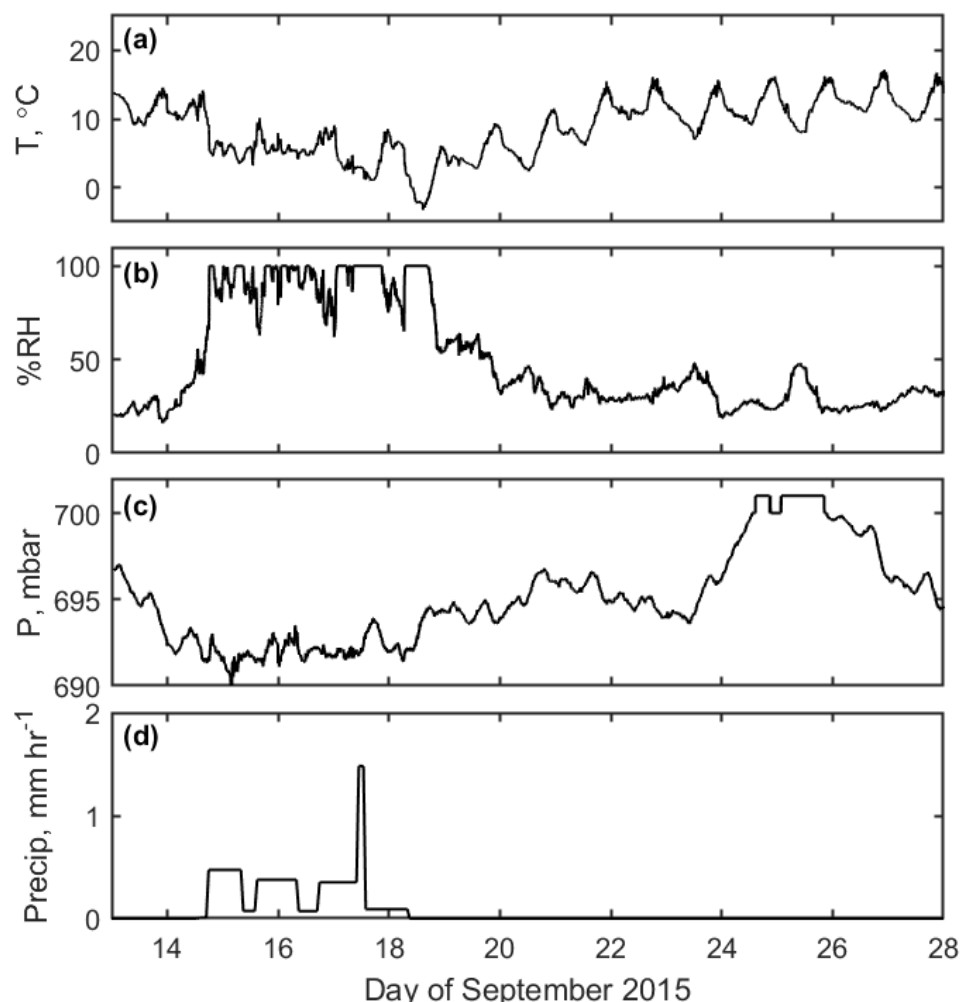


**Figure 1.** Weather conditions over the course of FIN-03, including (a) air temperature, (b) relative

humidity, (c) barometric pressure, and (d) precipitation rate.

Relatively warm, dry conditions were observed initially at the Storm Peak Laboratory.

Clear skies on September 11 and 12, 2015 gave way to clouds and haze on September 13.
Cooler temperatures, lower barometric pressure, and higher relative humidity (generally above >



70%) accompanied rainfall on September 14. This was followed by continued rain on September
15, intermittent rain and short periods of hail on September 16, a mixture of rain, snow, and sleet
on September 17, and snow on September 18. The next and longest period in the study,
September 19 to 28, was marked by an increase in temperature, an increase in barometric
pressure, lower relative humidity, and a lack of precipitation. More detailed weather records
including daily photographs and a summary of human-produced daily observations are
summarized in supplemental Section S1. Daily wind rose plots are provided in Figure S1.
**3.2 Aerosol context**
**3.2.1** *Aerosol size distribution and surface area*
The time series of aerosol size distribution measured by the LAS (in three hour means) is
shown in Figure 2a. The maximum and minimum total LAS concentrations were 706 cm$^{-3}$ and
74 cm$^{-3}$ respectively, and the mean and standard deviation of the total LAS concentration
throughout FIN-03 were 410 cm$^{-3}$ and 138 cm$^{-3}$, respectively. The highest total LAS
concentration recorded during FIN-03 (706 cm$^{-3}$) occurred in the early hours on September 25.
Elevated aerosol concentration (at least one standard deviation above the mean) was also
observed during midday on September 13, before and during midday on September 14, before
midday on September 25, in the afternoon on September 26, and around midday on September

27.

The timeline of LAS aerosol surface area in Figure 2b emphasizes that surface area was
predominately submicron throughout the study, with a mode at about 0.16 μm. This is important
to note, in combination with chemical composition information discussed in the next section
because it is relevant to understanding the likely sizes and surface areas of INPs. We will revisit
the surface area of INPs for use in parameterizations in a later section. Quantitative timelines of



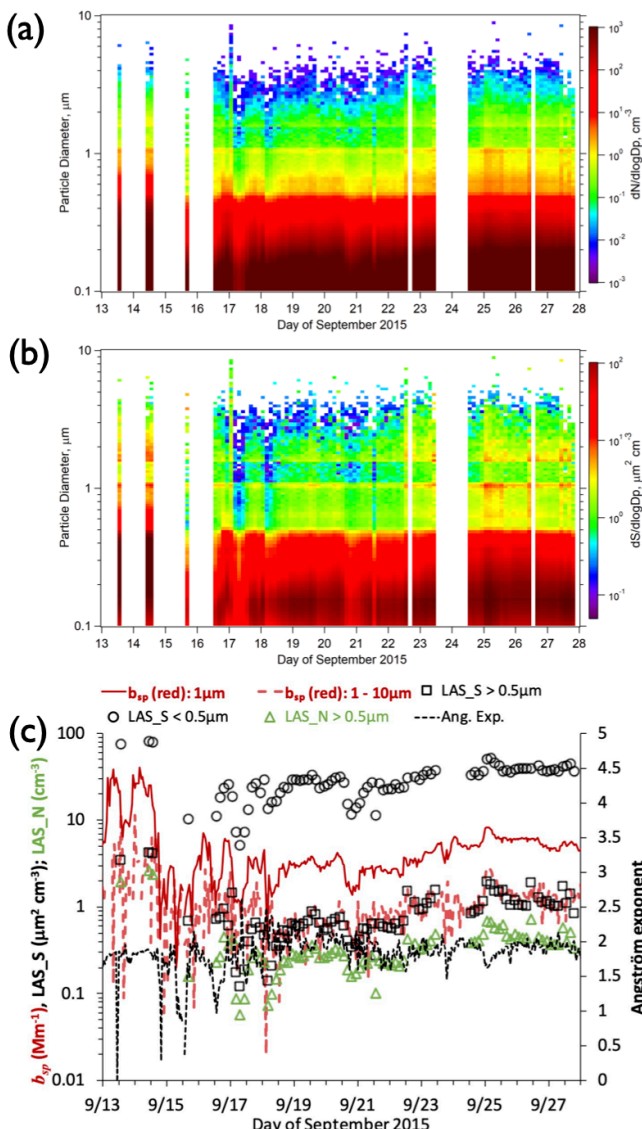

**Figure 2.** Time series of dry particle number concentration distribution (ambient conditions, not STP)

measured by the laser aerosol spectrometer (LAS) in a), shown as three-hour means at ambient pressure.

Time series of particle surface area distribution is in b). Timeline of nephelometer scattering (1-hr data) in

the red channel for < 1 μm and 1 - 10 μm size ranges, 3-hr LAS number concentration > 0.5 μm, 3-hr

LAS surface area at sizes below and above 0.5 μm, and Angström exponent (dashed, right axis).





LAS surface area above and below 0.5 μm are shown in Figure 2c. Surface area at above 0.5 μm
is about a factor of 30 lower than at below this size over most of the study period. Also shown in
Figure 2c is nephelometer scattering ($b_{sp}$) in the red channel (700 nm) showing a dominant
contribution when the upstream impactor was set to 1 μm (aerodynamic) and a much lower level
of 1 – 10 μm scattering. This scattering from coarse mode particles is consistent with and trends
with the LAS surface area in the supermicron regime, while the Angström exponent (calculated
using red and blue channels) being close to 2 (small particle dominance) throughout the study is
consistent with the dominance of submicron contributions to total surface area. Figure 2 also
emphasizes that the lowest aerosol concentrations and surface areas occurred during varied time
in the wet period of the study from midday on the 14[th] through the 17[th] of September.
**3.2.2 *Aerosol composition***

The number concentration of aerosol particles from 0.2 to 3 μm with characteristic

spectra belonging to eight composition categories (sulfate/organic/nitrate, biomass burning,
elemental carbon, sea salt, mineral dust, meteoric, alkali salt, and fuel oil combustion), and the
number concentration of unclassified aerosol particles by the PALMS, were assessed for three-
hour averages through the FIN-03 period. For simplicity, four of these categories (elemental
carbon, meteoric, alkali salt, and fuel oil combustion) were combined into a category called
"other" due to the low concentration of particles in each of these categories resulting in 6 total
classifications (categories (SulfOrgNit = sulfates/organics/nitrates, BioBurn = products of
biomass burning, Sea salt, Mineral dust, and Unclassified), as shown in Figure 3a. The three-
hour averages of the number fractions of each particle type were also calculated as the fraction of
the total aerosol number concentration measured by the PALMS in each of the six
classifications, as shown in Figure 3b.  The dominant categories throughout the FIN-03



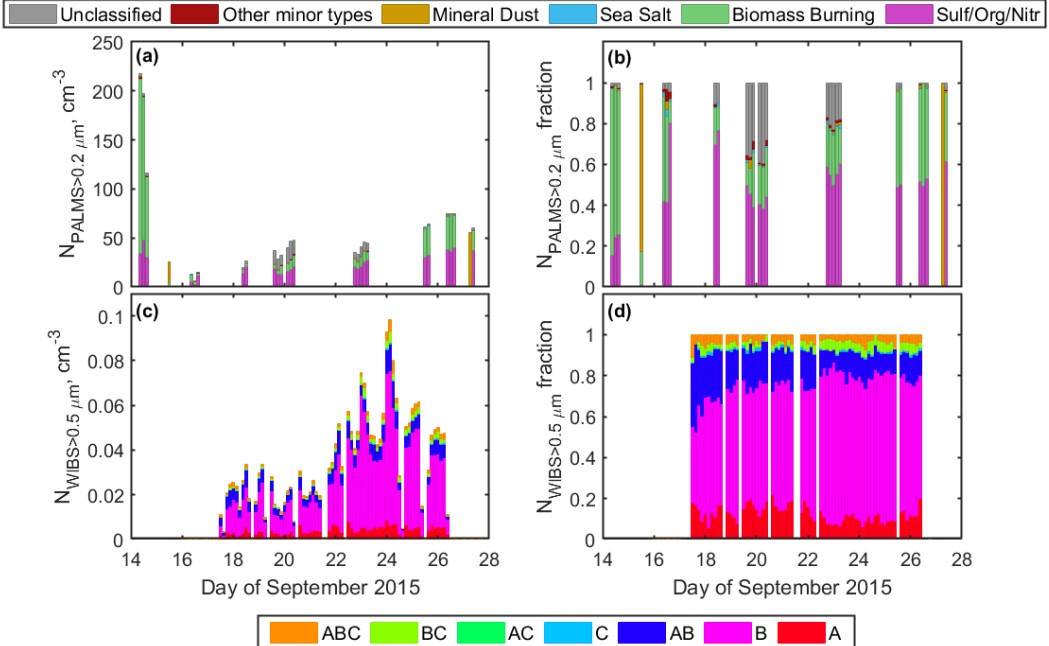

**Figure 3**. Subplots (a) and (b) show the aerosol particle number (ambient conditions, not STP) and

relative fractions (by cumulative count at all sizes) of each of the six PALMS compositional particle types

for the three-hour periods during which the PALMS was used to sample ambient air. Subplots (c) and (d)

show the aerosol particle number concentration and relative fractions (by count) of particles with diameter

> 0.5 µm in each of the channels (A, B, AB, C, AC, BC, and ABC, which are described in Perring et al.,

2015) over the course of the FIN-03 field campaign.



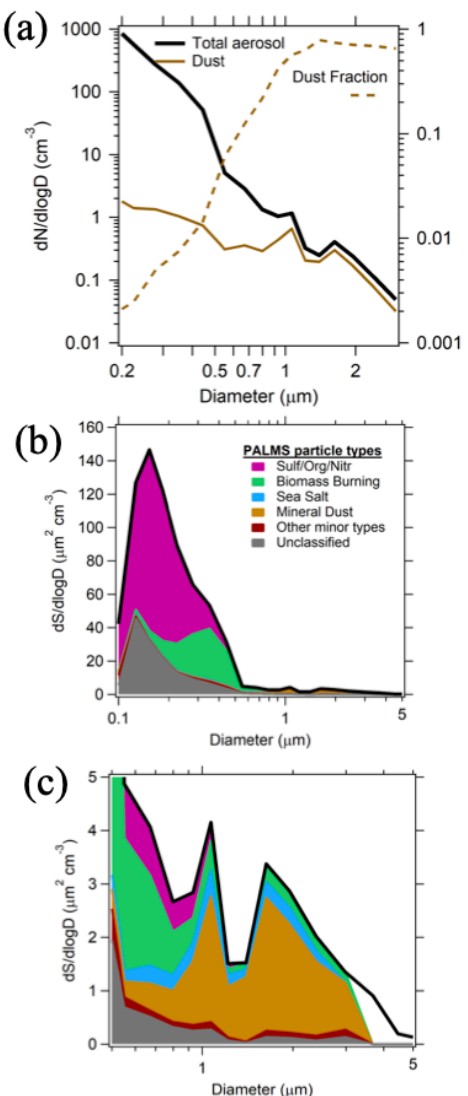

**Figure 4.** a) Total aerosol versus mineral/soil dust (ambient) number size distribution and dust fraction

interpreted from PALMS and LAS data for all times that the PALMS was sampling during FIN-03. b)

Surface area distribution differentiated for PALMS compositional types during the same sampling times.

c) Expanded plot from b) for the coarse mode size range to emphasize progressive dominance of dust

components at diameters > 0.5 μm.



campaign were BioBurn (mean 26 ± 43 cm$^{-3}$, maximum 177 cm$^{-3}$), SulfOrgNit (mean 22 ±13
cm$^{-3}$, maximum 48 cm$^{-3}$), and mineral dust (mean 3 ±11 cm$^{-3}$, maximum 55 cm$^{-3}$). The mineral
dust type also includes soil particles (crustal species mixed with organic material) (Zawadowicz
et al., 2019). The highest total particle number concentration measured by the PALMS (218
cm$^{-3}$) occurred on September 14 (of which 177 cm$^{-3}$ consisted of biomass burning and 34 cm$^{-3}$
consisted of sulfates/organics/nitrates). This biomass burning plume impacted the site for several
hours. Mineral/soil dust particles were ubiquitous throughout the study, with a concentration of
0.128 ± 0.446 cm$^{-3}$ (median and interquartile range). Anomalous concentrations >10 cm$^{-3}$
observed for a few 5-min sample periods are likely due to road dust emitted from site. Dust
concentrations were <1 cm$^{-3}$ for 90% of the PALMS samples. Mineral/soil dust represented a
median of 0.3% of particles in the >0.2 μm size range, increasing to 23% and 67% for >0.5 and
>1.0 μm particles (Figure 4a). Similarly, mineral dust contributions to total surface area are
inconsequential for total aerosol (Figure 4b) but dominate in the coarse mode regime for the
study (Figure 4c). We will revisit this result in discussions of parameterization of INPs in Section

3.5.

The daily average number concentration of fluorescing aerosol particles corresponding

with each of the seven WIBS-4A types with diameter > 0.5 μm is shown in Figure 3(c), and the
daily average number fraction of each WIBS-4A type is shown in Figure 3(d). The dominant
types of fluorescent aerosol particles throughout the FIN-03 field campaign were types B, AB,
and A, which on average accounted for 63.2%±8.7%, 16.0%±6.3%, and 12.5%±3.9% of the
particles detected by the WIBS respectively.

In contrast with the daily average number fraction in each PALMS category, the relative

contributions of each of the seven WIBS-4A particle types did not vary much over the course of





the study when the WIBS-4A was operational, with perhaps the exception that Type AB
decreased in prevalence from September 18 (42.9%) to September 21 (10.1%). A modest trend
from lower total fluorescing particle concentrations (0.02 to 0.04 cm$^{-3}$ at STP) through
September 21 to higher concentrations (0.07 to 0.15 cm$^{-3}$ at STP) from September 22 through
26$^{th}$. WIBS-4A data was not collected on September 13-16, nor on September 27. The first
period was somewhat critical to evaluating INP relations to bioaerosols, so we note here in
advance this caveat. Time-resolved size distributions for each WIBS-4A channel, as well as the
total particle concentration measured across these seven channels, are shown in supplemental
Figure S2. FBAP assignments related to INP predictions will be discussed in Section 3.5.
**3.3      Immersion freezing measurements**
A summary of the number concentrations of immersion freezing INPs ($N_{INP}$) over the
course of the field campaign, binned for one degree temperature intervals, is shown in Figure 5.
The concentration of INP detected over this range ranged over five orders of magnitude (0.01 to
160 L$^{-1}$). At any one temperature, differences up to a little more than one order of magnitude are
apparent in comparing average data from individual methods, mirroring results presented in
previous laboratory and field studies (Hiranuma et al., 2015; DeMott et al., 2017, 2018; Knopf et
al., 2021; Lacher et al., 2024).
As expected, a trend of increasing $N_{INP}$ with decreasing temperature was observed for the
FRIDGE-CS, CSU-IS, NC State-CS, and CSU-CFDC. The data represented by the single MIT-
SPIN processing temperature condition also falls well within the concentration range reported for
the other instruments. Incremental changes in $N_{INP}$ with decreasing temperature was similar for
all measurements that spanned a broad temperature range. This comparability of $dN_{INP}/dT$
contrasts with an apparent increasing high bias of drop suspension freezing measurements versus

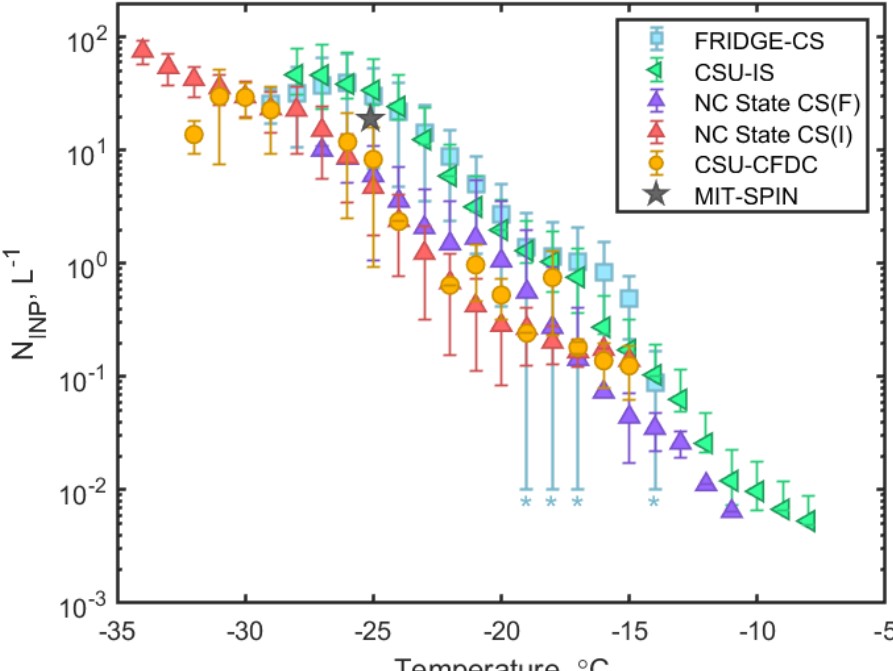


**Figure 5.** Campaign average immersion freezing INP concentrations (sL$^{-1}$) in 1 °C bins for instruments

participating in intercomparison studies.  Error bars represent one standard deviation in the measurement

means collected at the specified temperature and not measurement uncertainties. The star labels indicate

when these exceeded the mean for the FRIDGE-CS data. The times over which the INP concentration has

been averaged for each instrument is explained in the text.

CFDC measurements during comparable sampling at various surface sites (non-mountaintop or

free troposphere) found in DeMott et al. (2017) but agrees with FIN-02 laboratory studies

(DeMott et al., 2018) and recent atmospheric studies at Puy de Dome (Lacher et al., 2024). INP

concentration variability at single temperatures, reflected in Figure 5 as a standard deviation of

bin means, is likely due to variations in aerosol properties in response to production and

scavenging processes upstream of the site. Nevertheless, generally higher $N_{INP}$ measurements



744 were obtained with the FRIDGE-CS and the CSU-IS than the CSU-CFDC and NC State-CS (F)

745 and NC State-CS (I) analyses. Such biases in other studies have been attributed to different

746 efficiencies in sampling of largest particles (e.g., Lacher et al., 2024; Cornwell et al., 2023), but

747 the collection methods for offline measurements in this study were substantially similar, as

748 discussed further below. Hence, we cannot attribute measurement differences to a systematic

749 source. Comparability of impinger versus filter sampling methods for immersion freezing

750 measurements via the NC State-CS mirrors the findings in DeMott et al. (2017).

751   To compare the operation of these instruments over time, the mean and standard

752 deviation (when applicable) of immersion freezing $N_{INP}$ were calculated over three-hour periods

753 for each instrument (except for the MIT-SPIN) at 1 °C intervals (± 0.5 °C). Means are plotted as

754 a time series in Figure 6. Although some differences appear in comparing instrument by

755 instrument, as will be discussed, some general observations from the temporal data of Figure 6

756 are that INP concentrations at temperatures > –20 ºC were at a maximum during the precipitation

757 period, as might be expected for rainfall production of biological INPs (Huffman et al., 2013;

758 Mignani et al., 2021; Testa et al., 2021; Cornwell et al., 2023), while the strongest differences

759 between the concentrations of INPs active at higher and lower temperatures occurred during

760 period of warming under high pressure later in the study. The latter observation might be

761 expected for a strong contribution of dust-like INPs, with a steeper $dN_{INP}/dT$.

762   Periods of agreement and discrepancy are clearer in examining the ratios of time-matched

763 and temperature-matched three-hour immersion $N_{INP}$ values that were calculated for each pair of

764 instruments, as shown in Figure 7. As a positive note, the mean $N_{INP}$ reported by different

765 instruments for all temperature conditions taken together generally fell within a span of one order

766 of magnitude. Figure S3 shows the percent of immersion INP measurements in which all

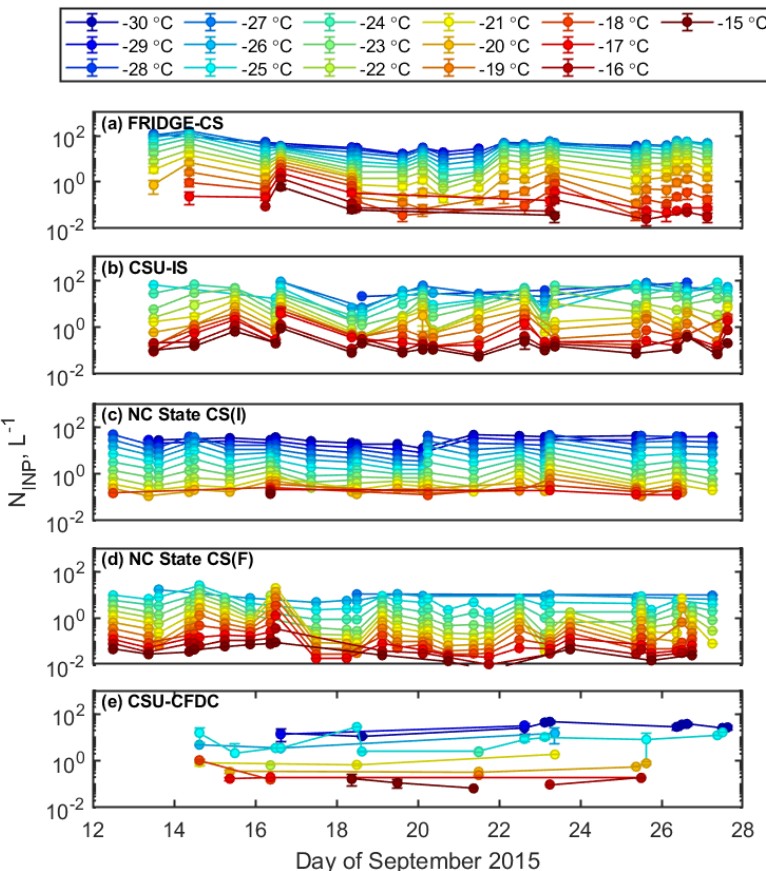

**Figure 6.** Time series of immersion-freezing mode INP concentrations (sL$^{-1}$) measured during

intercomparison periods by (a) the FRIDGE from Goethe University Frankfurt, (b) the CSU ice

spectrometer, (c) the NC State cold stage (collected using an impinger), (d) the NC State cold stage

(collected using the filter), and (e) the CSU continuous flow diffusion chamber. INP concentrations

shown in this figure are averaged over three-hour periods.



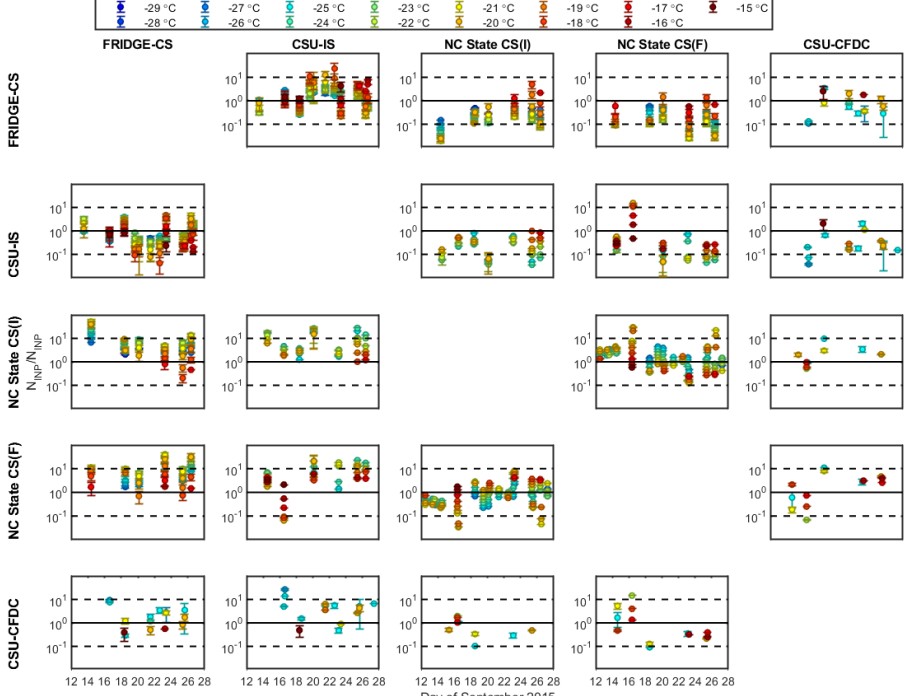

**Figure 7.** Ratios of the immersion freezing INP concentrations measured by each instrument, to the

immersion INP concentrations measured by each other instrument (three-hour averages). Each instrument

(FRIDGE, CSU-IS, NCSU-CS (I), NCSU-CS (F), and CSU-CFDC) is represented by one of the five

columns as well as one of the five rows.

instrument pairs agreed within one order of magnitude. This level of agreement compares well

with the findings from FIN-02, for which the immersion $N_{INP}$ measured by several online and

offline instruments agreed within an order of magnitude. This is encouraging given that FIN-02

was a laboratory intercomparison on single composition aerosol samples consisting of particles

with diameter < 2 μm whereas FIN-03 was a field campaign in which constant changes in the

concentration, size distribution, and composition of the ambient aerosol population at Storm





Peak Laboratory were all potential factors.  This shows that field data can be collected with
nearly the same level of accuracy as laboratory experiments. While also mimicking the results of
DeMott et al. (2017) for a smaller instrument comparison exercise, agreement was slightly
poorer than found in another recent intercomparison where INP concentrations were stated to
match within a factor of 5 (Lacher et al., 2024).

Reiterating what is apparent from campaign-wide results in Figure 5, the best agreement

for short-term periods throughout the study as shown in Figure 7 was observed between the
FRIDGE-CS and the CSU-IS, in which only one out of seventy-two 3-hour, time- and
temperature-matched $N_{INP}$ (1.4%) did not agree within an order of magnitude. Nevertheless,
discrepancies of a few to several times did occur from September 19th onward, focused more
often at >–22°C, with the CSU-IS measuring higher from the 19th to the 22nd and the FRIDGE-
CS higher at some other times, notably the 23rd and 26th of September. None of these periods
were distinguished in any discernible manner by weather or aerosol properties. For example,
LAS and PALMS concentrations were no more than 20% different than the FIN-03 campaign
means during any of these periods. Aerosol surface areas were about a factor of two lower
overall during the 19th to 22nd period than for the period after the 23rd (Figure 2), which does not
imply a special sampling bias for larger particles for the IS filter that was open to the air, a point
we will discuss further below.

Both the FRIDGE-CS and CSU-IS showed high bias from a few to more than 10 times

versus NC State-CS(I) or CS(F), primarily at processing temperatures below –20 °C, whereas
agreement was generally very much better at >–20 °C. The poorest agreement overall was
observed for the CSU-IS compared to the NC State-CS(I), a combination for which 12 out of 44
(27%) immersion $N_{INP}$ means did not agree within an order of magnitude.  Agreement between



809 the FRIDGE-CS and the NC State-CS(I) was only slightly better, as 13 out of 52 (25%) time-

810 matched $N_{INP}$ means did not agree within an order of magnitude. Higher than order of magnitude

811 such discrepancies at lower temperatures were markedly present on September 14, 23 and 26.

812 Based on PALMS data, the 14th was richer in compounds from biomass burning, poorer in

813 sulfates, organics, and nitrates, and slightly poorer in mineral dust than average, as discussed in

814 Section 3.2. The concentration of > 0.5 μm particles measured by the LAS during this time was

815 also relatively high (2.5 cm$^{-3}$ compared to the campaign mean 0.45±0.62 cm$^{-3}$). However, the

816 14th is not markedly distinguished overall in the timeline of all INP measurements in Figure 6, so

817 perturbations to composition and concentrations of all particle sizes due to the biomass burning

818 event did not appear to specially perturb the INP populations. We have already noted that the

819 23rd and 26th of September had aerosol populations that were not much different than the project

820 mean on those days.

821  The CSU-CFDC INP measurements generally agreed with the other measurements within

822 an order of magnitude for data collected on the same day and temperature, and its measurements

823 of INP concentration were in best agreement with all methods for temperatures > –20 °C. CSU-

824 CFDC INP concentrations tended to be lower than those from the FRIDGE-CS and CSU-IS at

825 temperatures below –20 °C. A similar divergence in online versus offline $N_{INP}$ measurements in

826 this temperature range was reported by DeMott et al (2017) for ground-based sampling, with

827 online measurements tending to measure progressively lower INPs than offline integrated filter

828 or impinger collections at below –20 °C, approaching one order of magnitude below –25 °C. At

829 the Puy de Dome mountain station (Lacher et al. 2024), only modest and insignificant

830 underestimates were made by the CSU-CFDC (again, with a 2.5 μm impactor) versus offline

831 INP concentrations when all were measured from a PM10 inlet. CSU-CFDC INP measurements





were comparable on average with measurements from the NC State-CS(I) and NC State-CS(F),
consistent with the mean results shown in Figure 5. Comparing the timeline of ratios of NC
State-CS(I) to NC State-CS(F), only 5 out of 130 (4%) of the INP concentrations obtained
through analysis by the identical off-line apparatus differed by more than an order of magnitude.

A possible explanation for INP measurement discrepancies that has been tendered in

other intercomparison campaigns sampling ambient air is that INPs are highly sensitive to the
size range of collected aerosol, and systematic size-dependent differences in collection
efficiencies vary for different collection types (DeMott et al., 2017; Knopf et al., 2021; Lacher et
al., 2024). For example, Lacher et al. (2024) found significant underestimates of INPs by both
online and offline methods measuring from the PM10 inlet versus offline measurements from
filter collections made on the laboratory rooftop. In this study, as we have noted above, a
similarly consistent difference between rooftop versus laboratory or between online and offline
measurements is not found. FRIDGE-CS measurements from the turbulent-flow inlet and CSU-
IS measurements from the rooftop filter agreed on average over the course of the study. CFDC
INP measurements agreed reasonably well with the NC State (F) and (I) measurements. Larger
particles do tend to have higher ice nucleation efficiency, so biases in their collection can lead to
sometimes large differences in assessed INP concentrations (Mason et al., 2016). Disaggregation
of the very largest collected particles when placed in water suspensions has also been implicated
for discrepancies between different substrate collections (DeMott et al., 2017; Lacher et al.,
2024). An obvious size-based collection bias existed for the online INP instruments, which had
impactors upstream to limit particles >2.5 μm (50% cut-size) from entering. There may have
been additional line losses for these instruments sampling from an inlet and using tubing to
transfer particles, though these tend to be of minor influence at below the impactor size cut





(Knopf et al., 2021). The impinger is known to be less efficient for small (<200 nm) and large
(>10 µm) particle capture, but unless the relatively light to moderate wind conditions at the inlet
during FIN-03 conferred some special bias, Hader et al. (2014) predict a 50% capture efficiency
at near 10 µm. The filter samplers on the rooftop should have been equivalent, with the only
difference in the orientation of filters for the NC State samples being mounted face-down. The
size bias in this configuration is unknown. The FRIDGE filter should have captured particles
with the same efficiency as the turbulent flow inlet, since only a very short line connected the
filter to the interior inlet structure in the laboratory. Only if very large INPs > 13 µm were
dominant by number amongst total INPs, which is unexpected, would the FRIDGE filter
collection have been expected to differ from the rooftop filter collections.

In the end, it seems more likely that unquantifiable random and non-random sources of

discrepancy, related to such things as sample size, instrument temperature sensor drift, varied
instrument cooling rates and inconsistency in sample materials or handling and storage (e.g.,
Barry et al., 2021b; Beall et al., 2021), may also contribute to the fact that measurements of
immersion freezing INP concentrations from ambient air are generally uncertain by up to one
order of magnitude, as this study once again supports.
**3.4 Relation of immersion freezing INPs to aerosol properties**

While establishing correlations between INPs and aerosol properties were not a focus of

the intercomparison, the ancillary aerosol data did allow for inspecting some simple linear
correlation analysis. This provides insight into the size range of greatest relevance for the INP
intercomparison period. Throughout the campaign, a positive and significant trend between total
LAS particle concentration (i.e., > 0.1 µm) and $N_{INP}$ was observed for FRIDGE-CS (R = 0.55-
0.74 and p < 0.05 for measurements at –28 °C < T < –15 °C), but no clear statistically significant





trend was observed between total LAS particle concentration and $N_{INP}$ for the other four
instruments (Figure S4a). A greater number of significant positive trends were found between the
concentration of particles with diameter > 0.5 μm and $N_{INP}$. This was the case for the FRIDGE-
CS (R = 0.54-0.94 and p < 0.05 for measurements at –28 °C < T < –19 °C), CSU IS (R = 0.46-
0.72 and p < 0.05 for measurements at –21 to –25 °C), NC State CS(I) (R = 0.46-0.61 and p <
0.05 for measurements at –29°C < T < –24 °C), and the NC State CS(F) (R = 0.51-0.64 and p <
0.05 for measurements at –26 °C < T < –22 °C).

No consistent, significant (p < 0.05) correlation was found between changes in

composition (from the PALMS categories and WIBS-4A types) and immersion freezing $N_{INP}$
across the range of setpoint temperatures employed during FIN-03 (Figure S4b).

**3.5 Inferences to INP compositions during FIN-03**

To provide context for the discussed intercomparisons and because this study provides

data needed for testing the relevance of existing parameterizations of ice nucleation in regional
and global climate models (Andreae & Rosenfeld, 2008; Morris et al., 2011; Seifert et al., 2011),
we utilize some previously-developed ice nucleation parameterizations for specific compositions
to diagnose consistency or not with INP compositions in the high altitude environment of FIN-
03. We examine parameterizations for mineral dust INPs that have different links to larger size
particle concentrations (DeMott et al., 2015) versus mineral dust surface area (Niemand et al.,
2012), and biological INPs as linked to fluorescent particle concentrations (Tobo et al., 2013;
Twohy et al., 2016). Hereafter we will refer to these parameterizations as DeMott 2015,
Niemand 2012, and Tobo 2013. We also utilize a more direct method of probing INP
compositions using the IS sample treatments discussed in Section 2.2.2.



Each of the above-noted deterministic parameterizations was used to predict $N_{INP}$ at –30

°C, –25 °C, –20 °C, and –15 °C using the equations and inputs described in Table 2 and
summarized below. We do not attempt an analysis using stochastic parameterizations.
1)  DeMott 2015 is based on CSU-CFDC laboratory measurements of ice nucleation on

mineral dust soil samples as well as field data from situations dominated by mineral dusts

(i.e., dust plumes from major deserts), collected for CFDC operational conditions

essentially the same as for this study (i.e., simulated immersion freezing conditions at

105% RH) (DeMott et al., 2015). For FIN-03, aerosol concentrations measured by the

LAS (> 0.5 µm dry diameter) and converted to STP concentrations were used as the input

for this parameterization for comparison to INP data that is also reported at STP

concentrations. Predictions also depend on temperature (Table 2). Since PALMS data

indicates that dust particles dominated the coarse mode only at sizes above 1 µm in

diameter (Figure 4), we first adjust LAS data accordingly for the percentage of dust

particles with diameters > 0.5 µm as input to this parameterization, which we have

already stated is 23%. A correction factor (CF) of 3 was also applied (as indicated in

Table 2) according to the results in DeMott et al. (2015) which showed that when

applying the parameterization to represent immersion freezing INP concentrations in a

model or in comparison to other immersion freezing methods, this CF is needed to

account for CFDC underestimates of immersion freezing INPs (see Methods). The CF is

applied in this case because calculations will be compared to the average $N_{INP}$ from all

measurements.

2)  The Niemand 2012 parameterization (Table 2) for mineral dust INPs is based entirely

from laboratory measurements and incorporates measurements of temperature and





923  particle surface area as the basis for prediction of INPs. It is especially important to limit

924  the size range of aerosols for which this parameterization is applied, because total surface

925  area was dominated by small particles in FIN-03. Therefore, with reference to Figure 4,

926  we will assume that all dust surface area occurs at sizes larger than 0.5 μm and represents

927  50% of that surface area.

928 **Table 2** Summary of INP parameterizations.

| Parameterization | Equation | Constants |
|---|---|---|
| Mineral dust INPs: Niemand et al. (2012) | $$N_{INP}(T_C) \approx n_s(T_C)S_{tot} = (a\ exp\ (b(T_C)+c))(S_{tot})$$ $N_{INP}(T_C)$ = INP concentration (sL-1) at T (Celcius) | a = 1x10⁻⁹ b = -0.515 c = 8.821 |
| Mineral dust: DeMott et al. (2015) | $$N_{INP}(T_K) = (cf)(n_{a>0.5\mu m})^{(\alpha(273.16-T_K)+\beta)}$$ $$exp\ (\gamma(273.16-T_K)+\delta)$$ $N_{INP}(T_K)$ = INP concentration (sL⁻¹) at T (Kelvin) $n_{a>0.5\mu m}$= mineral particle number concentration > 0.5 μm (scm⁻³) cf = 1 (CFDC data comparison) or 3 (other immersion freezing) | $\alpha = -0.074$ $\beta = 3.8$ $\gamma = 0.414$ $\delta = -9.671$ |
| Fluorescing biological aerosol particle INPs: Tobo et al. (2013) | $$N_{INP}(T_k) = (N_{FBAP>0.5\mu m})^{(\alpha'(273.16-T_k)+\beta')}$$ $$exp\ (\gamma'(273.16-T_k)+\delta')$$ $N_{INP}$ = INP concentration (sL⁻¹) $N_{FBAP}$ = FBAP concentration (scm⁻³) | $\alpha'$ $= -0.108$ $\beta' = 3.8$ $\gamma' = 0$ $\delta' = 4.605$ |
| Fluorescing biological aerosol particle INPs: Cornwell et al. (2023) | $$N_{INP}(T_C) = f(T_C)1000N_{FBAP>0.5\mu m}$$ f(T_C = -20 °C) = 0.318 f(T_C = -15 °C) = 0.016 | N/A |

930 3) As discussed earlier, we use two definitions of FBAP at sizes larger than 0.5 μm to and

931  temperature to predict biological INP concentrations based on Tobo 2013 as defined in

932  Section 2.1, presuming to bracket low and high estimates of their links to INPs. We also



explore links of higher temperature freezing data (> –20 °C) to FP3 particles, using the

same scalings of the relation between FP3 concentrations and INP concentrations as a

function of temperature that were established by Cornwell et al. (2023) for a coastal

California environment. While we have no reason to expect that these scaling factors

listed in Table 2 are valid for the high altitude, continental environment of FIN-03, they

are starting points to explore this additional link of certain FBAP particles to INPs.

To compare these parameterized values with observations, an overall mean observed

immersion freezing $N_{INP}$ was calculated for each three-hour period based on all the available data
from all the instruments. This was considered as a reasonable approach since it factors in the
inherent variability found between methods. Immersion freezing $N_{INP}$ was predicted for each
parameterization using mean WIBS-4A, and LAS data, both at STP concentrations, collected in
the coincident 3-hour periods of time as the INP data. The observed and predicted immersion
freezing $N_{INP}$ are plotted against each other in Figure 8. Four temperatures of comparison (–15, –
20, –25 and –30 °C) are presented in Figure 8 for DeMott 2015, Niemand 2012, and Tobo 2013,
while two temperatures of comparison (–15, –20 °C) are used for links to FP3-based prediction
of biological INPs. Temperatures are indicated via levels of shading of the data points.
Using the constraint on mineral particles from the combination of PALMS and LAS data for the
campaign average, predicted INPs underestimate the mean INP concentrations at all
temperatures (Figure 8a). The Niemand 2012 surface-area-based INP estimates come modestly
closer to observations, averaging 25% of the total INP concentrations for all times and all
temperatures, while the DeMott 2015 predictions average 4% of INP concentrations, with large
variability apparent. These results can be expected to be highly sensitive to the assessed average

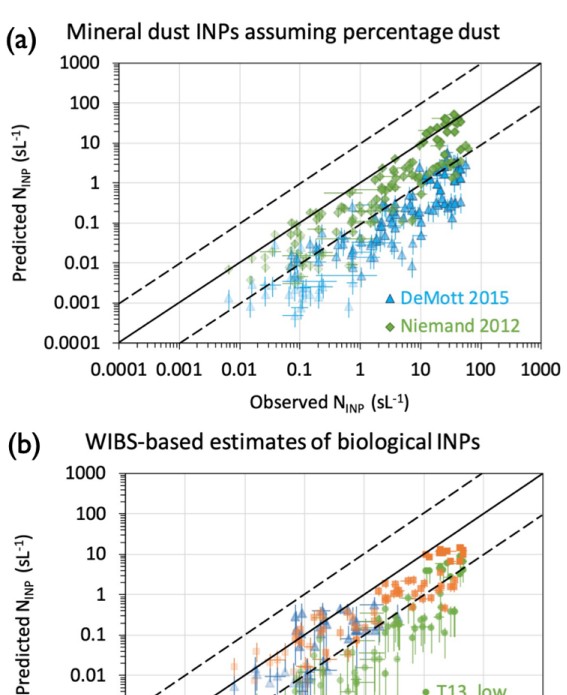


**Figure 8**. a) Comparison of mean observed $N_{INP}$ (all instrument average) and predicted $N_{INP}$ calculated

from DeMott et al. (2015) (DeMott 2015) and Niemand et al. (2012) (Niemand 2012) mineral dust INP

parameterizations at temperatures -30 °C, -25 °C, -20 °C, and -15 °C (gradations in shading from dark to

light) for the PALMS estimated percentages of dust particle number and surface area at sizes above 0.5

μm. Mean $N_{INP}$ are averaged over three-hour periods and plotted uncertainties are standard deviations.

Predicted $N_{INP}$ uncertainties are propagated based on 25 % uncertainty in aerosol number and surface area

concentrations. b) Comparison of mean observed $N_{INP}$ and predicted $N_{INP}$ calculated from

parameterizations linking to FBAP concentrations from *Tobo et al.* (2013) (T13_low and T13_high; see

text for description) and from *Cornwell et al.* (2023) (C23_FP3) following the FP3 particle definition of

*Wright et al.* (2014). Only -15 and -20 °C comparisons are shown for the FP3 prediction. The solid line in

each plot is the 1:1 line and the dashed lines represent an order of magnitude in both directions.





mineral particle fraction at sizes above 0.5 µm (varied over the study) and on whether particles
that have a source from regional soils will be represented only by those with mineral content.
Therefore, for comparison, parameterization results using an assumption that all particles at
diameters exceeding 0.5 µm were dust particles are presented in Figure S5. In this case,
admittedly a somewhat unrealistic maximum assumption on mineral dust numbers and surface
area, Niemand 2012 estimates a dust source for 50% and DeMott 2015 estimates 25% of
observed INPs on average. Thus, the predictions of the two parameterizations become more
closely aligned for assumption of more overall mineral dust particles in the size range larger than
0.5 µm. Discrepancy has been noted previously in applying these parameterizations to link to the
aerosol model in an Earth System model for the Southern Ocean region (McCluskey et al.,
2023). In that case, calculations were based on aerosol model derived dust distributions and
occurred under very low dust loading scenarios where neither parameterization has been firmly
tested in the laboratory or field. Under both assumptions on mineral particle number, since
DeMott 2015 was developed based on CFDC measurements for particles < 2.5 µm in the field
and laboratory, a low bias compared to Niemand 2012 might be expected in comparison to
average immersion freezing data that includes larger particles.

The timeline of predicted $N_{INP}$ for the two dust parameterizations in comparison to mean

observed $N_{INP}$ is shown in Figure 9 for the same temperatures used in Figure 8. These analyses
emphasize that 1) INP observations do not show a special enhancement during the biomass
burning event at the start of FIN-03, and hence closer agreement of the dust parameterizations
with observations at that time is likely an artifact of attributing dust-like INP activation
properties to the dominant biomass burning compositions at that time; 2) the predicted $N_{INP}$
trends better with observed $N_{INP}$ at temperatures < –20 °C, as expected for a dominance of dust-
like INPs; and 3) the predictions fare less well in describing the observed INP populations at > –
20 °C where biological INPs may be expected to have greater influence. Thus, these analyses
overall suggest the presence of a dust-like immersion freezing INP type during FIN-03, but that
the typical INP efficiency (INP as a function of dust concentration and temperature) attributed to
mineral dust underestimates the freezing behavior of INPs overall during the period of study.

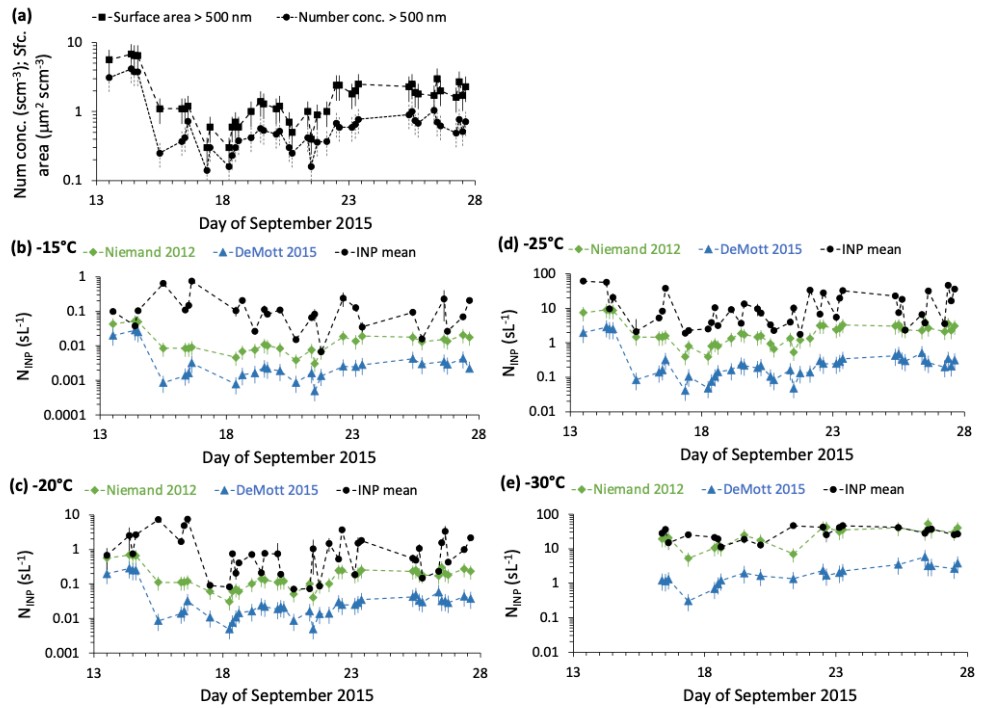


**Figure 9.** Time series of aerosol number concentration and surface area (3-h averages at STP) in a), and
observed mean measured immersion freezing $N_{INP}$ (INP mean) plotted with predicted $N_{INP}$ from the
mineral dust parameterizations of Niemand 2012 and DeMott 2015 as described in the main text (all
three-hour averages at STP) at temperatures of -15, -20, -25, and -30 °C in b) to e), respectively. Dashed
lines are intended only to connect data points and do not imply knowledge of intermediate values.
Uncertainties mark one standard deviation above and below the mean values of all parameters.

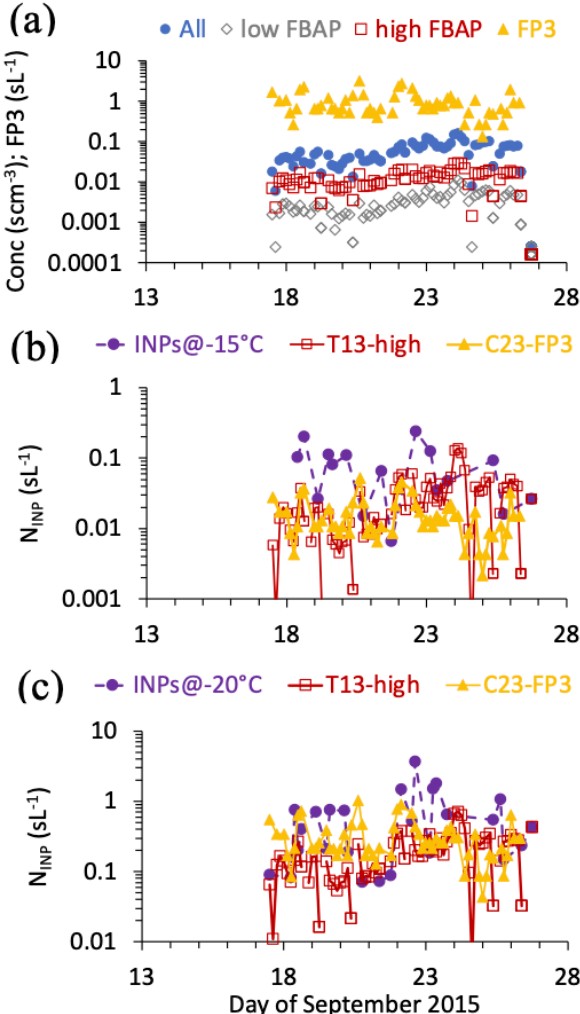

**Figure 10**. a) Timelines of WIBS-based fluorescent particles assignments (all fluorescing in any channel, low and high FBAP, and FP3 particles), as defined in the text, during FIN-03. b) INP observed mean concentrations and biological INP parameterization predictions linked to high FBAP following Tobo et al. (2013) (T13-high) and FP3 particles following Cornwell et al. (2023) at -15 °C in b) and -20 °C in c).





For FIN-03, the Tobo parameterization of biological INPs consistently underpredicted

$N_{INP}$, independent of the WIBS FBAP definition used, denoted as T13_low and T13 high in
scatterplot comparison of measured versus predicted values at all times and temperatures in
Figure 8b and the timeline comparisons at –15 and –20 °C shown in Figure 10. Figure 10 also
shows the timeline WIBS total fluorescent particle concentrations, the high and low FBAP
concentrations, and FP3 concentrations. The higher FBAP prediction of INPs falls much closer
to the observations than the low FBAP prediction in Figure 8b and shares some proximal
equivalence to observations at –15 to –20 °C at times. This result is like that found by Twohy et
al. (2016) for air over the site where Tobo et al. (2013) collected their data, with the higher
FBAP estimate bounding the upper end of measured immersion freezing INP concentrations at
temperatures > –20 °C. Also notable in Figure 8b and Figure 10 is that the C13-FP3 INP
concentration predictions filled a similar space as the T13_high estimates, coming closest
together at –20 °C. While these results suggest that biological INP parameterizations can explain
the higher temperature INP concentrations observed during FIN-03, with caveats on the large
and likely not fully quantifiable uncertainty in such predictions, the temporal analysis
(Figure S6) indicates that there is no consistent temporal agreement between predicted and
measured INPs, even if different scaling factors were applied to the predictions. Predictions at
–20 °C again show the best overall agreement, while those at –15 °C suggest that the Cornwell et
al. (2023) scaling factor should be higher for the SPL site at the time of FIN-03 to better describe
mean values of biological INP concentrations using the FP3 particle signal.



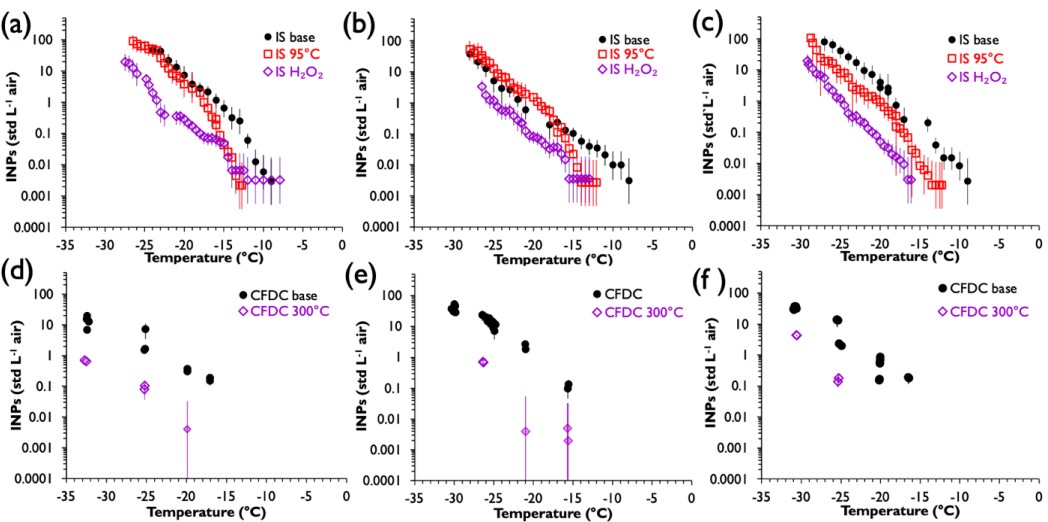


**Figure 11.** Summary of treated IS filter suspensions using heat and peroxide (a, b, c) and dry heat-treated

CSU CFDC single particle data (d, e, f), for September 15, 23 and 25 (a-c, d-f, respectively). Error bars

represent 95% confidence intervals for individual experimental spectra for the CSU-IS and for individual

CSU CFDC measurements.

        The results of CSU-IS and CSU-CFDC treatments on INP concentrations measured for

three (of 21 overall) intercomparison time periods are shown in Figure 11, for examination of

consistency with the results of the diagnostic parameterization analysis just discussed. In Figure

11a-c, it is seen that thermal treatments indicated the strong contribution of inferred biological

INPs primarily at temperatures higher than about –20 °C, but that peroxide digestion of organic

compounds lowered INP activity at all tested temperatures by an order of magnitude on average.

Similar reductions of INPs measured for single particles by the CSU-CFDC following dry

heating (Figure 11d-e) demonstrate strong consistency with the IS results for bulk immersion

freezing on the dominance of organic INP compositions, even though CSU-CFDC measured

unamended INP concentrations were always lower. The CSU-IS heat treatment results (Figure



11a-c) suggest that biological INPs may have been ubiquitous during FIN-03 at temperatures
above –20 °C, and extended to lower temperatures at times, as indicated by the results from
September 25. This is broadly consistent with the parameterization results based on FBAP
measurements, although the Tobo 2013 and FP3 parameterizations did not capture all the
influence of apparent biological INPs during the study. Whether for size-limited (< 2.5 µm) as in
CSU-CFDC measurements, or bulk aerosol collected for CSU-IS immersion freezing
measurements, the inferred INP compositions typically dominated by organics at temperatures <
–20 °C could reflect origins from arable soil dusts (Testa et al., 2021) that surround the region of
study. Biomass burning aerosols also have influence as organic INPs (Schill et al, 2020; Barry et
al., 2021). However, while biomass burning type particles were noted as a prevalent composition
in FIN-03, these types of potential INPs likely cannot explain INP concentrations in FIN-03
because Barry et al. (2021) showed that Western U.S. biomass burning INPs have active site
densities about 3 orders of magnitude lower than those attributed to dust particles that also were
ubiquitous at modest number concentrations during FIN-03. Furthermore, the strong biomass
burning event noted on September 14 had only modest, if any, apparent impacts on INP
concentrations despite greatly elevated aerosol concentrations and surface areas, as already
mentioned above (Figure 9).

Finally, in Figure 12 we address whether the treatment results support the conclusion of

the diagnostic parameterization analysis suggesting that inorganic INPs (mineral particles in
particular) were of minor influence during FIN-03. We introduce the additional parameterization
of Tobo et al. (2014) (Tobo 2014) for arable soil dust INPs as part of this discussion. Tobo et al.
(2014) parameterized the ice nucleation behavior of soil dusts from Wyoming, regionally
proximal to the FIN-03 site at SPL, specifically using the CSU CFDC and the dry heat method at



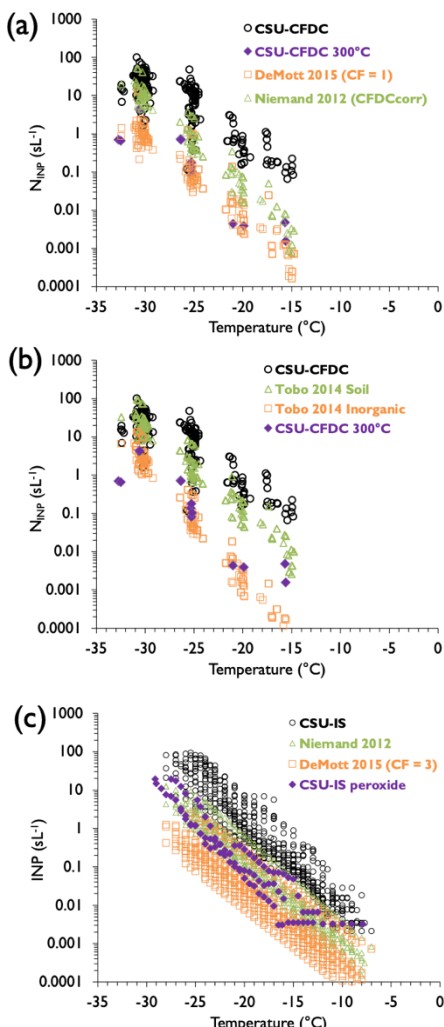


**Figure 12**. a) Comparison of all untreated CSU CFDC data (black circles), cases after passing through the

upstream 300 °C tube heater (purple diamonds), and calculations from the DeMott 2015 dust

parameterization in (orange squares) and with CF = 1 as appropriate for a direct comparison to CSU

CFDC data (see text). b) The same exercise as in a) but using predictions of total soil organic INP

concentrations and inorganic INP concentrations within soil INPs, both from Tobo et al. (2014). c) The

same exercise but for all CSU-IS data and the cases with peroxide digestion. In this case, CF = 3 must be

used in DeMott 2015 and the mineral dust INP prediction of Niemand 2012 is also shown.





300 °C to indicate organic versus inorganic INP contributions from such soil particles. This
parameterization, like Niemand 2012, is based on the surface area of dust particles and so we
apply the same assumptions as before to restrict to the proportion of dust larger than 0.5 µm.
Since the CSU-CFDC is also restricted to measuring INPs at diameters below 2.5 µm, we apply
a correction factor to the surface area to account for the fact that the surface area at below this
size was 90% of the project average total surface area. No significant impact of the treatments is
assumed on aerosol concentrations or surface area at sizes above 0.5 µm in Figure 12.

Figures 12a and 12b focus on specific comparisons to CSU-CFDC data. In Figure 12a, it

is seen that INP concentrations predicted by the DeMott 2015 parameterization for sampling
periods during the entire campaign show remarkable agreement with the 300 °C CSU-CFDC
data on selected days when applying CF = 1 in the parameterization, as is appropriate for a direct
comparison to instrument data that is uncorrected for the underestimates that led to selecting CF
= 3 for modeling studies. In Figure 12 b, it is shown that the Tobo 2014 parameterizations for
untreated soil dusts and the inorganic remnants also give very good agreement with CFDC
untreated and treated data, supporting the likely important influence of such arable soil dusts
during FIN-03. Predictions for untreated soils do not quite reach the level of the observed INPs,
but this could be explained by the additional contribution of biological INPs that has already
been discussed.

In Figure 12c, direct comparisons of the Niemand 2012 and DeMott 2015 predictions for

mineral dust INPs for the entire project are shown in comparison to the CSU-IS untreated and
$H_2O_2$ treated data on selected days. The DeMott 2015 prediction of INP concentrations uses CF
= 3 in this case, as appropriate. The same discrepancy between the DeMott 2015 and Niemand
2012 predictions as discussed already regarding Figure 8a appears in this comparison.




Nevertheless, it is seen that both parameterizations grossly underestimate untreated CSU-IS INP
concentrations and the treated CSU-IS results fall between the predicted values, agreeing better
with the Niemand 2012 parameterization. While one might wish to allude to the fact that the IS
filters sample particle sizes, to 10 µm and possible larger that may have higher ice nucleation
efficiencies, while the CSU-CFDC was restricted to sampling particles <2.5 µm as a source for
the lower DeMott 2015 estimate in comparison to CSU-IS data, we have already addressed that
there was no general consistency in INP concentrations for methods that sampled similar size
particles overall. The best that can be stated is that the parameterization exercises and treatment
data strongly support that inorganic INPs were of weak influence during FIN-03 and that arable
soil dusts and biological INPs accounted for the strongest influences during sampling, akin to the
findings of Testa et al. (2021).
**3.6 Observations of INPs in the deposition nucleation regime**
Measurements of deposition nucleation $N_{INP}$ are summarized in Figures 13 and 14.
FRIDGE-DEP nucleation substrates were collected for 1 to 5 periods on many days during
FIN-03 and processed at 5-degree interval temperatures from –15 to –30 °C, and for setpoint
humidity of 95% and 99% RH (uncertainties to 2%). Data collected at 102% via the standard
FRIDGE methods are not included herein. CSU-CFDC and MIT-SPIN deposition data were
collected nominally at 95% RH with an uncertainty of about 2.5% RH, and at a range of
temperatures on different days.  Mean values and standard deviation error bars of the FRIDGE-
DEP data are shown in Figure 13a and median values of FRIDGE-DEP $N_{INP}$ (with interquartile
values as error bars) are shown in Figure 13b. Standard deviations were large over the course of
the study for comprehensive FRIDGE-DEP data when binned at 5-degree interval temperatures.
Nevertheless, average concentrations of deposition INPs measured by the FRIDGE-DEP



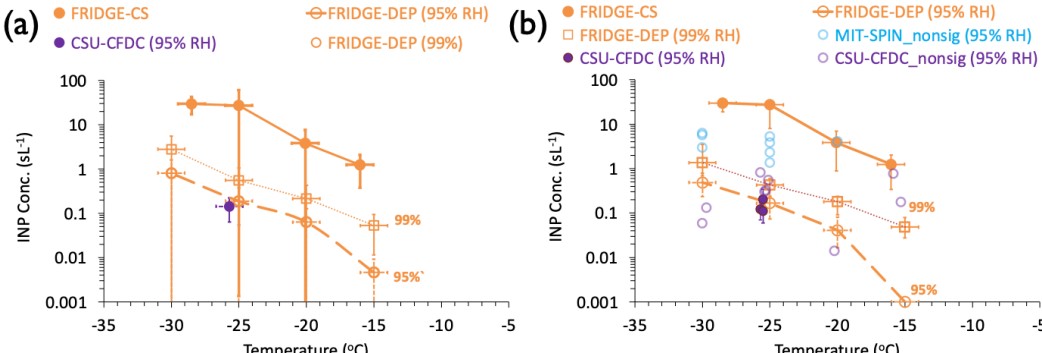

**Figure 13.** Summary of deposition-mode $N_{INP}$ (sL$^{-1}$) as a function of temperature. In a), mean FRIDGE-DEP data at 95% (open orange circles) and 99% (open orange squares) RH are shown along with mean immersion freezing data from the FRIDGE-CS (filled orange circles) and the mean for the few cases of statistically significant CSU-CFDC data (filled purple circle) at 95% RH. Error bars are one standard deviation of the means. In b), median FRIDGE-DEP data are shown and error bars for these are the 95% confidence intervals. The significant CSU-CFDC measurement points at 95% RH are also shown with their 95% confidence intervals. Data measured at 95% RH from the CSU-CFDC and MIT-SPIN that were positively valued but failed significance testing are shown without errors as open purple and open blue circles, respectively.

indicated a consistent 3-5 factor increase between 95 and 99% RH over the range of temperatures investigated. $N_{INP}$ concentration differences at the two RH values were slightly smaller for median values (Figure 13b), and the median values are slightly lower than the means. Finally, FRIDGE-CS values are plotted in each panel of Figure 13, indicating that FRIDGE-DEP $N_{INP}$ concentrations averaged for 99% RH are factors 10 to 30 lower than average immersion freezing $N_{INP}$ concentrations, depending on temperature.





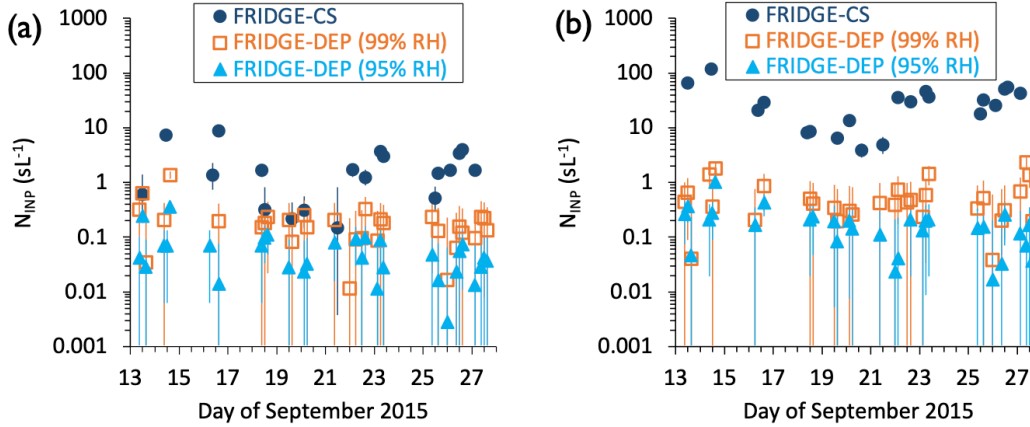


**Figure 14**. Time series of FRIDGE-CS (immersion freezing) and FRIDGE-DEP (deposition) $N_{INP}$
measured at a) -20 °C, and b) -25 °C. Data are from individual filters or wafer collections and error bars
are 95% confidence intervals.

One day of significant data was obtained for the CSU-CFDC deposition measurements

while using the aerosol concentrator, on September 14, containing three different time periods.
These are averaged to create the only online data point represented as a mean in Figure 13a. The
individual period measurements from this day, with confidence intervals as errors, are shown for
the CSU-CFDC in Figure 13b. Thereby it is seen that these measurements at close to –25 °C
agree very well with the mean FRIDGE deposition $N_{INP}$ at –25 °C and 95% RH. No
measurements of significance were achieved with the MIT-SPIN when operating in the
deposition regime. In fact, the most common CSU-CFDC and MIT-SPIN deposition nucleation
$N_{INP}$ results were below instrument detection limits, not meeting the test for significance, as
shown for all periods from 6 common days of such observations represented in Figure 13b.
Understanding that these data represent a failure to collect statistically-defensible data, the non-
significant data generally scatter about the significant CSU-CFDC data and the FRIDGE-DEP



data at 95% RH, with a higher bias for the MIT-SPIN data. This indicates the difficulty for
online continuous flow instruments to capture low deposition $N_{INP}$ concentration data that fall
below 1 sL$^{-1}$ at most times, considering the FRIDGE-DEP data as the standard. Higher sample
volumes and limited background frost conditions are needed to sense these low atmospheric INP
concentrations.

Time series of the FRIDGE-DEP measurements at –20 °C and –25 °C are shown in

Figure 14. Deposition-mode $N_{INP}$ has been averaged over three-hour periods for this analysis.
The FRIDGE immersion freezing data is included in this figure to allow for direct comparison
temporally. Immersion freezing $N_{INP}$ generally exceeded deposition-mode $N_{INP}$ when both types
of measurements were collected by the two FRIDGE operational methods within the same period
(or during adjacent time periods).  This difference ranged from 0 to 2 orders of magnitude, with
the largest differences seen at –25 °C and a period of insignificant differences between the mode
results seen only from the 18[th] to the 22[nd] of September at –20 °C (Figure 14a).

Based on these FRIDGE-CS and FRIDGE-DEP results, immersion-mode ice nucleation

dominates at most times at mixed-phase cloud temperatures. Nevertheless, deposition-mode ice
nucleation contributes modestly to the pool of INP at mixed-phase cloud temperatures in the
atmosphere, and thus may bear consideration for parameterization in atmospheric models. The
ability of online ice nucleation instruments to measure $N_{INP}$ in the deposition mode in
correspondence to offline measurements has not been confirmed due to the mentioned inability
of the online instruments used in FIN-03 to capture the low deposition nucleation $N_{INP}$
concentrations. More work should be carried out on measurements of INPs in the deposition
mode to understand variabilities in time and their relation to INP size and composition, as well as
to resolve if online measurements can be improved. For the time being, the substrate methods



appear to be recommended for ambient atmospheric measurements in the realm below water
saturation at mixed-phase cloud temperatures.

**4. Summary and conclusions**
FIN-03 was an ice nucleation instrument intercomparison conducted in the challenging
environment of the high-altitude mountaintop field setting. Two online systems (CSU-CFDC,
MIT-SPIN) and three offline systems (FRIDGE, CSU-IS, NC State-CS) were represented in
FIN-03. The immersion freezing INP concentrations measured in FIN-03 spanned a dynamic
range of over five orders of magnitude ($10^{-3}$ to $\sim 10^2$ L$^{-1}$) over the temperature range –35 °C to –5
°C. Agreement within one order of magnitude in immersion freezing $N_{INP}$ was generally
observed between all ice nucleation instruments measuring immersion INP concentrations at any
given temperature if measurement and sampling times were matched to within 3 hours. Better
than one order of magnitude agreement was found at temperatures lower than –25°C and higher
than –18°C, with occasional deviations larger than an order of magnitude in the temperature
range –25 °C to –18 °C. We do not have a full understanding of what controls better or worse
agreement at different times or different temperatures, though some factors have been previously
discussed in documenting FIN-02 laboratory studies (DeMott et al., 2018). In this study, there
was some inference that the different filters and impinger used did not equally capture particles
in all size ranges, which is something to improve on in future studies. Given the constant changes
in the concentration, size distribution and composition of the ambient aerosol population,
inevitable with any field campaign, this level of agreement represents state-of-the-art, at least as
judged based on recent laboratory and other field comparisons using similar instrumentation
(e.g., Knopf et al., 2021; Lacher et al., 2024).





Although FIN-03 was not conducted as a closure study per se, ancillary data on aerosol

sizes and compositions as recommended in more recent discussions of needs for true closure
exercises (Knopf et al., 2021; Burrows et al., 2022) were purposefully collected for integration
into analyses. This included explicit measurements of the aerosol size distribution, and single
particle measurements of aerosol chemical and biological composition. These measurements
allowed inferences to be made about INP compositions that provide context for the period of
study and establish an example for future intercomparison and long-term measurement efforts.
Through comparing INP data to some current parameterizations describing biological, mineral
and soil dust INPs, and additional direct investigations of INP composition via certain pre-
treatments to remove biological and organic immersion-freezing INPs, these investigations
revealed ubiquitous biological and organic-influenced soil-dust-like INP influences that mimic
those found over other continental regions (Knopf et al., 2021; Testa et al., 2021; Lacher et al.,
2024). Biological INPs were indicated via selected immersion freezing heat treatments to be
dominant at > –20 °C, although of potential influence at all mixed-phase temperatures.
Prediction of these based on parameterizations that utilize single particle fluorescence data (Tobo
et al., 2013; Wright et al., 2014; Cornwell et al., 2023) suggest the average utility of such
parameterizations but these were unable to predict the full temporal variation of biological INPs.
This suggests that local variations of these INPs, which may in fact represent multiple biological
particle types, is an area that requires more effort. Based on relatively good consistency between
predicted and measured mineral influences on immersion-freezing $N_{INP}$ concentrations, strictly
mineral or other inorganic components of INPs were suggested to have a modest contribution to
total INP concentrations at most times and at the freezing temperatures probed during this study.
As in most prior studies, the mineral influence became stronger at the lowest temperatures





assessed. In contrast, it was found by comparison to a parameterization based on proximally
regional soil particles that arable soil INPs likely explained the second most important
contribution of INPs during FIN-03, those emanating from other organic particle components
that may have been internally mixed with minerals. Biomass burning influences were possible
but appear to have not contributed greatly to the climatology of INPs during the study. It was
critically important in arriving at these conclusions to have single particle aerosol composition
data, from a mass spectrometer that could discern the sizes and fractional contribution of
minerals and from a laser-based single particle fluorescence measurement to estimate the
biological character of particles. Nevertheless, a great amount of work is still needed to generally
parameterize the mixed INP populations that may occur temporally in the atmosphere at higher
altitude sites like SPL, or anywhere for that matter.

Importantly, FIN-03 included an assessment of the separate relative contributions of

deposition and immersion freezing INP concentrations, one of the few existing data sets of this
kind. The offline FRIDGE-DEP method was used to acquire comprehensive deposition $N_{INP}$
measurements in dependence on RH (95 and 99%), while the CSU-CFDC and MIT-SPIN
instruments attempted focused deposition nucleation measurements at (nominally) 95% RH on
several days. As expected, FRIDGE-DEP measurements indicated factor of a few increases in
deposition $N_{INP}$ concentrations between 95 to 99% RH. Also, deposition $N_{INP}$ concentrations
were nearly always lower than immersion freezing $N_{INP}$ concentrations. Deposition INP
concentrations at most times at 99% RH (always at 95% RH) were lower by an order of
magnitude than immersion freezing INP concentrations at –20 °C and by more than an order of
magnitude at –25 °C. For the online instruments, only limited periods of deposition INP
measurements on one day achieved statistical significance from the CSU-CFDC data. While



these data were in good agreement with FRIDGE-DEP data at –25 °C and 95% RH, the most
striking result was that all other measurement periods for the CSU-CFDC and MIT-SPIN gave
measurements that were not significant at the 95% confidence level. Thus, currently, offline
methods for measuring deposition INPs appear to offer the best chance for success in measuring
the lower concentrations of INPs that activate below water saturation in the mixed-phase
temperature regime. It would be useful to make such assessments at a variety of sites to confirm
measurements made during FIN-03 on the relative contributions and variability of INPs active in
these conditions toward ice formation in clouds. Additional instrument developments for online
measurements of these, and future intercomparisons, will be useful.

In summary, the relative agreements amongst instruments during FIN-03 that match those

found in the FIN-02 laboratory studies are encouraging and represent steady improvement in the
community's collective ability to detect and quantify atmospheric ice nucleation. There was not a
clear divide between the ability of online and offline systems to measure INP concentrations
from the data collected in this study, although the need to carefully consider aerosol sampling
efficiencies for different instruments was highlighted as a potential issue in this study, and one
requiring close attention in future studies. In principle, both types of instruments show excellent
promise for future field studies. For full closure studies of ice nucleation by atmospheric
aerosols, methods for identifying INP composition as demonstrated herein and recommended by
other recent discussions in Knopf et al. (2021) and Burrows et al. (2022) are critical for
understanding and improving INP measurements overall.



**Data availability** All data used for the figures in this paper can be accessed at
https://radar.kit.edu/radar/en/dataset/eGhfvcOhsOyADZXN (persistent
doi:10.35097/eGhfvcOhsOyADZXN)
**Author contributions**
Paul J. DeMott, Jessica A. Mirrielees and Sarah D. Brooks wrote the paper with assistance from
all teams and authors contributing information on instrument descriptions and comments on all
results and conclusions, with contributions from Jake Zenker on some data analysis. Paul J.
DeMott, Ezra J.T. Levin, Thea Schiebel, Kaitlyn Suski, and Tom Hill provided data and analyses
from the CSU-CFDC and IS instruments. Daniel J. Cziczo, Martin J. Wolfe, Sarvesh Garimella,
and Maria Zawadowicz provided MIT-SPIN team measurements and analyses. Markus D.
Petters and Sarah S. Petters provided data and analysis for the NC State-CS instrument. Heinz G.
Bingemer, Jann Schrod, and Daniel Weber provided data and analyses for the FRIDGE
instrument. Anne Perring provided data and analyses for the WIBS-4A. Karl Froyd provided
data and analyses for the LAS and PALMS. Anna Gannet Hallar and Ian McCubbin oversaw
field operations, coordinated with visiting teams at Storm Peak Laboratory, and provided
nephelometer and meteorological measurements. Paul J. DeMott, Daniel J. Cziczo, Ottmar
Möhler contributed to organize the campaign in connection with the other FIN activities.

**Competing interests**
The contact author has declared that none of the authors has any competing interests.
**Acknowledgements**



Partial financial support for this project was provided by the U.S. National Science
Foundation, Grant No. AGS-1339264 and U.S. Department of Energy's Atmospheric System
Research, an Office of Science, Office of Biological and Environmental Research program,
under grant no. DE-SC0014487. Paul J. DeMott, Ezra J.T. Levin, Thea Schiebel, Kaitlyn Suski,
and Tom Hill acknowledge partial and in-kind research support during FIN-03 from NSF grant
no. AGS-1358495. Markus Petters acknowledges partial and in-kind support during FIN-03 from
NSF grant no. AGS-1450690. Jann Schrod acknowledges research support from the European
Union's Seventh Framework Programme (FP7/2007-2013) project BACCHUS under grant
agreement no. 603445. Heinz G. Bingemer and Daniel Weber acknowledge research support
under DFG grant BI 462/3-2.  Thea Schiebel and Ottmar Möhler received support through the
German Science Foundation Projects INUIT and INUIT-2 (MO 668/4-1 and MO 668/4-2). Anne
Perring acknowledges support from the NOAA Health of the Atmosphere Program and the
NOAA Atmospheric Composition and Climate Program. Special thanks to Romy Fösig (Ullrich)
for assistance with data archival.




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
