# Peer review of "Field Intercomparison of Ice Nucleation Measurements: The Fifth International Workshop on Ice Nucleation Phase 3 (FIN-03)"

_EGUsphere, 2024_

## Referee Comment (RC2)

**Review of "The Fifth International Workshop on Ice Nucleation Phase 3 (FIN-03): Field Intercomparison of Ice Nucleation Measurements," by P. J. DeMott et al. 2024**

It was a pleasure to read "The Fifth International Workshop on Ice Nucleation Phase 3 (FIN-03): Field Intercomparison of Ice Nucleation Measurements," by P. J. DeMott et al. 2024. The manuscript presents data and summarizes results from the third of a series of instrument measurement comparisons that were aimed at achieving better collective community understanding when it comes to measuring ice nucleating particles (INPs). The focus of the third effort was utilizing a subset of instruments used in the first two phases, to focus on making field measurements of ambient air. While many of the same instruments were used in FIN-02[1], that study's focus was on laboratory-based measurements. The field-based FIN-03 measurements add complexity due to the heterogeneous nature of ambient aerosol and the low naturally occurring concentrations of INPs. The FIN campaigns are well known and brought much of the community together, but given the summarized measurements were conducted in 2015 it has been a bit of a wait to see the outcomes of the 3rd phase published. The results summarized in the submission remain well worth reporting and valuable to the community at large. However, in its current form the manuscript lacks polish and consistency, which diminish its readability. Moreover, given the large gap between when the measurements were made and their submission in this form, the submitted work seems to, in places, lack connections to work published in the intervening years. This is not a blanket statement, as some more contemporary results (references) are used, but there is a lack of consistency that again points to a need for a systematic review of the manuscript.

I suggest that the author's take into consideration the contextualized comments I include below and resubmit an updated manuscript.

**Itemized Scientific and Editorial Comments:**

*Suggestions are given by line number taken from the downloaded pre-print PDF document:*

• (Abstract, 48) strike "a subset of"

• (Abstract, 50) suggest a rephrasing to: Composition of the total aerosol was characterized using ...

• (Abstract, 55) suggest a rephrasing to: Mineral dust containing particles were ...

• (Abstract, 56)  should probably be diameters

• (Abstract, 58)  Here and throughout the text (lines 93, 186, 187 etc.) the symbol $\sim$ is used where $\approx$ would be a better choice. Although not always strictly followed, generally the former, 'similar to', means the same order of magnitude. Most often the author's intention it seems is the later, which is 'approximately'.

• (Abstract, 67)  Should this say: ....order of magnitude or more, more efficient

• (Abstract, 76) strike "at most times"

• (79) suggest a rephrasing to: Aerosol particles that ...

• (140-143) A reference (and probably in other places) to Brasseur et al.[2], seems like it should be included. Much like the campaign summarized by the Lacher et al.[3] paper, the Brasseur et al.[2] Measurement Report describes intercomparison and measurements by instruments for counting INPs, including a subset of the instruments used in FIN-03.

• (171) suggest to replace "aerosol" with particle.

• (228) A Thomson et al. (2000) paper is referred to but is not found in the listed references.

• (323-324) suggest a rephrasing to: ...all ice nucleation instruments utilized in FIN-03 is provided in Table 1. Detailed operating principles, locations of samplers ....

• (362) "Frost corrections are defined through..." The wording is strange here. Suggest replacing through with, using or utilizing, or somehow rephrasing.

• (§2.2) A general comment on the presentation of instruments and instrumental setups in §2.2, both §2.2.1 and §2.2.2. I would suggest the author's do a careful re-reading of the instrument sections

and that they re-write the sections such that equivalent information is presented for each instrument. During reading the lack of consistency became most apparent in §2.2.2, but this forced me to return to the earlier text and notice a general lack of consistency. When presenting different instruments different details are presented, and it is not evident why or what, if any, importance these differences indicate. For example, for the NC State - CS the temperatures of sample storage are included, but not in the case of the CSU-IS. My intent is not that the author's should include all the information for all the instruments. Rather the author's should decide on what are the important and relevant parameters and make sure to present them uniformly. The reader should have the same information for all apparatuses. Alternatively, they must tell the reader why some things are important for some instruments but not for others.

The current presentation is both a-systematic and simultaneously uninformative as to why/how the information that is presented is chosen.

• (370) "(by 3 time)" by a factor of 3?

• (378 and again at 407) The o.d., which I take to mean 'outer diameter' is an irrelevant dimension here, as it gives no information concerning inner diameter.

• (411-412) The "low-pass filter" sentence seemingly comes from nowhere? Why are these counts removed? I do not follow the logic at the beginning of this paragraph.

• (447) suggest: ...has been previously...

• (462) suggest: ...except that Teflon tape replaced stopcock grease sealing the impinger...

• (491) suggest: ...2 m distant.

• (501-502) suggest: ...suspension estimated using Eq. 1. (Vali has been cited when presenting the equation.)

• (549-550) suggest: ....specific analytical procedures differ.

• (560) suggest: ...is derived using Eqs. 1 and 2.

• (563-566) This is a confusing description of the EAC's working architecture and may not even be necessary given the instrument papers available. I would suggest re-writing if it is to be included. For example, what does "12 kV against a grounded..." mean? I think that it must be that the 12 gold wires have an applied 12 kV electric potential.

• (576) suggest: At SPL samples were taken with the EAC for ...

• (Figure 2, plus caption) The font in panels (a) and (b) is extremely small and hard to read, make uniform with (c) which is much better. (c) is entirely missing from the caption, although I take it that it is described beginning with, "Timeline...."

• (666) extra (

• (759-760) suggest: ...during a period of warming

• (766) I find it strange that figure S3 is referred to before figure 7 appears in the text, as S3 seems to be a distillation of Figure 7. It seems the authors should carefully consider their choices as it regards these results and figures. Moreover, both aforementioned figures strictly speaking present the 1:1 comparison incorrectly. In Figure S3 a blue FRIDGE-CS bar should show 100% agreement in the FRIDGE-CS column, and so on for all instruments compared with themselves. Currently they are missing, and the caption does not mention that the 1:1 comparison is ignored because it is 100%. The same is true of the diagonal of the figure matrix in Figure 7. In fact, S3 is a bit of a strange presentation, and perhaps some sort of matrix like presentation with a heat map (for example) would be better, but "agreement within 1 order of magnitude" is a bit awkward in general. Figure 7 is more informative but quite busy, and this might be the better figure for the supplement. Albeit I would include 1:1 or 100% agreement along the diagonal for rigor.

In general, I would suggest this information could be better presented and communicated and would suggest the author's re-imagine these figures.

• (Figure 6) Are all temperatures necessary? For the main text perhaps a distilled figure focusing on a few temperatures, or results for temperature averages could be a better alternative and more effective at communicating the point. Also, some lines connect points across missing data points at the warmest temperatures, where I assume no activation was observed. I would suggest that these points are better left unconnected, for clarity of the figure, but also because it is likely the value is actually 0 INP or below the resolution. For the CSU-CFDC panel (e) such data points are not connected, which is a much better approach than in some of the upper panels.

In the figure caption, use the accronyms as they were introduced in the text. Strike "from Goethe University Frankfurt". This kind of information appears in the text and does not need to be reproduced in a caption.

• (page 40) A general question when using the comparison that 1 order of magnitude is a good agreement. It would be worthwhile to present a simple calculation to quantify how many particles (i.e., activated droplets or volumes in immersion freezing) result in an order of magnitude change. At some temperatures and concentrations, the counting statistics may be rather poor, it would be interesting to address this given the already dilute nature of the ambient sampling.

• (Table 2) I see the CF constant discussed in the text, but no discussion of the other constants in the Table. Are these taken from previously published literature? Or used as fitting parameters here in some minimization scheme? Please illuminate.

• (989) "trends better" what is meant here? correlates?

• (1200) again the Brasseur et al.[2], seems like it should be included.

• (1240-1241) Rephrase, "factor of a few increases" Something here is awkward and unclear. Deposition freezing increases with increasing RH?

• (1246) Again awkward phrasing, "...achieved statistical significance from the CSU-CFDC data."

• (References) In addition to the previously mentioned Thomson text, the Burrows, 2022 paper that is referred to is not found in the Reference list.

—

[1] DeMott, P. J., Möhler, O., Cziczo, D. J., Hiranuma, N., Petters, M. D., Petters, S. S., Belosi, F., Bingemer, H. G., Brooks, S. D., Budke, C., Burkert-Kohn, M., Collier, K. N., Danielczok, A., Eppers, O., Felgitsch, L., Garimella, S., Grothe, H., Herenz, P., Hill, T. C. J., Höhler, K., Kanji, Z. A., Kiselev, A., Koop, T., Kristensen, T. B., Krüger, K., Kulkarni, G., Levin, E. J. T., Murray, B. J., Nicosia, A., O'Sullivan, D., Peckhaus, A., Polen, M. J., Price, H. C., Reicher, N., Rothenberg, D. A., Rudich, Y., Santachiara, G., Schiebel, T., Schrod, J., Seifried, T. M., Stratmann, F., Sullivan, R. C., Suski, K. J., Szakáll, M., Taylor, H. P., Ullrich, R., Vergara-Temprado, J., Wagner, R., Whale, T. F., Weber, D., Welti, A., Wilson, T. W., Wolf, M. J., and Zenker, J. (2018). The Fifth International Workshop on Ice Nucleation phase 2 (FIN-02): laboratory intercomparison of ice nucleation measurements. *Atmospheric Measurement Techniques*, 11(11):6231–6257.

[2] Brasseur, Z., Castarède, D., Thomson, E. S., Adams, M. P., Drossaart van Dusseldorp, S., Heikkilä, P., Korhonen, K., Lampilahti, J., Paramonov, M., Schneider, J., Vogel, F., Wu, Y., Abbatt, J. P. D., Atanasova, N. S., Bamford, D. H., Bertozzi, B., Boyer, M., Brus, D., Daily, M. I., Fösig, R., Gute, E., Harrison, A. D., Hietala, P., Höhler, K., Kanji, Z. A., Keskinen, J., Lacher, L., Lampimäki, M., Levula, J., Manninen, A., Nadolny, J., Peltola, M., Porter, G. C. E., Poutanen, P., Proske, U., Schorr, T., Silas Umo, N., Stenszky, J., Virtanen, A., Moisseev, D., Kulmala, M., Murray, B. J., Petäjä, T., Möhler, O., and Duplissy, J. (2022). Measurement report: Introduction to the HyICE-2018 campaign for measurements of ice-nucleating particles and instrument inter-comparison in the Hyytiälä boreal forest. *Atmospheric Chemistry and Physics*, 22(8):5117–5145.

[3] Lacher, L., Adams, M. P., Barry, K., Bertozzi, B., Bingemer, H., Boffo, C., Bras, Y., Büttner, N., Castarede, D., Cziczo, D. J., DeMott, P. J., Fösig, R., Goodell, M., Höhler, K., Hill, T. C. J., Jentzsch, C., Ladino, L. A., Levin, E. J. T., Mertes, S., Möhler, O., Moore, K. A., Murray, B. J., Nadolny, J., Pfeuffer, T., Picard, D., Ramírez-Romero, C., Ribeiro, M., Richter, S., Schrod, J., Sellegri, K., Stratmann, F., Swanson, B. E., Thomson, E. S., Wex, H., Wolf, M. J., and Freney, E. (2024). The Puy de Dôme ICe Nucleation Intercomparison Campaign (PICNIC): comparison between online and offline methods in ambient air. *Atmospheric Chemistry and Physics*, 24(4):2651–2678.

---

## Referee Comment (RC3)

Referee comment by Gabor Vali on the "Fifth International Workshop .... " by DeMott and co-authors.

This manuscript reports the results of a workshop held in 2015 at the Storm Peak laboratory in Colorado with a large collection of ice nucleation and aerosol instruments. The main goals were to assess the degree of agreement among ice nucleation instruments of various designs and to interpret the results with consideration of aerosol characteristics. These goals have been mostly achieved. The agreements found are significant and the authors also make clear where discrepancies appeared. The results are described in detail.

Since the process of ice nucleation is complex, inferences about how it takes place in the atmosphere or in the laboratory have been arrived at over many decades from the way the process is manifested in instruments that proceed from prior condition to the appearance of ice with controlled characteristics. There are inherent limitations to this approach as knowledge is hard to achieve of relevant aerosol properties and of the way conditions are altered on the way till observation of ice becomes possible. Yet, by gradual understanding of the impacts of instrument design approaches have been key to the gradual development of a degree of understanding of how to achieve usable measurements of heterogeneous ice nucleation by atmospheric aerosol particles and by some types generated from known substances.

The workshop described in this paper undertook to test how instruments which have been shown to yield agreements among themselves with controlled aerosols will perform in the relatively clean high-altitude environment sampling ambient aerosol. Similar efforts have been reported previously, but this paper reports on significantly more extensive measurements. No surprises emerged from taking the devices to high elevations and, perhaps, that can be taken as reassurance of their reliability.

The paper is long and demanding. Instruments and sampling setups are detailed, possibly to a greater extent than necessary since previous publications are available for all. On the other hand, the results are presented in a more compressed style. The aerosol data are used to interpret outliers in the intercomparisons but this is somewhat cursory. Going beyond the intercomparisons, testing agreements with assumed particle compositions and parameterizations (Section 3.5) could well have been a separate paper. That section and the aerosol data needed for it take up, by a rough estimate, about 1/3 of the paper. Similarly, the deposition mode data, from only one instrument, could be separated from the main subject.

There is little attention in the paper on the absolute values of INP concentrations observed. The mountain-top observatory could be expected to yield INP concentrations lower than those observed a low altitudes. This doesn't appear to be the case.

The main result of the intercomparisons of instruments is stated as an order of magnitude agreement for immersion freezing INPs. This is a good result from the perspective of instrument reliability compared to, say, what the situation was 20 years ago. On the other hand it is worth seeing it as a 5ºC discrepancy for given INP concentration. For atmospheric cloud processes that is still a huge uncertainty. Thus, for INP measurements to be useful predictors of cloud evolution at given location and times there is much remains to be accomplished. This does not diminish the accomplishments demonstrated in the paper, but it may be worth reflecting on in the Abstract and in the Conclusions.

On its own terms, the evaluation of prior parameterizations with the extensive aerosol data available from FIN-03 is a laudable idea. Combined with similar efforts a better appreciation is developed for the validity of the generally used predictive equations for atmospheric INP concentrations.

Essentially all data reported is for measurement temperatures of -15°C and lower. This limitation should probably made clear to the readers right up front. The reasons for this limitation also deserve to be stated. A

further caveat applies to having all comparisons done in terms of the cumulative number concentration, with only brief attention to the slopes of the temperature spectra.

The degree of agreement among instruments is extensively analyzed with reference to Figs. 6 and 7. Another striking point that can be added to that analysis that while FRIDGE-CS and CSU-IS show significant changes with time between Sept 14 and 16, specially at the higher temperatures, these changes are not evident in the other data. Then, for the next two days, these two instrument show greater disagreement in Fig. 7.

In the title, placing importance on what was done, rather than on the vehicle for doing it would be useful, i.e. "Field Intercomparison of Ice Nucleation Measurements; the Fifth ....."

Line numbers:

77-176    The second part of the Introduction is not as clear as it should be for readers not already familiar with the subject. More specific comments are given in the following.

120    The text of the preceding lines calls for a somewhat more detailed reference to history. Ice nucleation comparison workshops go back as far as 1967. The main reason for many repetitions is not a difference in goals but the development of new instruments and better characterization of the processes underlying the different instruments, i.e. better understanding of heterogeneous ice nucleation. Another important factor was growing recognition of having detailed aerosol characterization (lines 169-172) and better instruments for that information. All of these factors see advances, but important limitations also remain.

121-150    Following on the previous comment, it would be helpful to readers to have a concise argument for why FIN-01 had to be followed by -02 and 03. Why was -03 deemed necessary, The contrast with Knopf et al (2021) and Lacher et al (2024) would be best detailed at the end of characterizing FIN-03 goals.

210-215    This may not be as simple as it appears here. Not all sizes of aerosol are of the same composition even at a given instant of time and the INPs are a very small subset of the total aerosol, so the manner in which aerosol composition is correlated with the INP measurements needs detailed description and substantial caveats. The subject should perhaps be addressed after the description of the instrumentation.

235    " ... positive or negative mass spectra"  - what does that mean?

236-241    This helps to clarify what was meant on lines 210-215, but only in part. Are the sample sizes of the two instruments the same?? If  PALMS samples only a fraction of the particles sampled by LAS, how much uncertainty enters in the analyses?

Section 2.1 The authors' approach to present how data from given instruments are interpreted along with the description of the instrument has merits. Caveats and limitations are stated up front. However, it separates those issues (some listed in the foregoing) from the actual interpretation of the results. In the end, I think it isn't working to the paper's advantage.

313    "All these instruments ..."  refers to the types of devices or the specific units?

326    Please clarify " ..substantial temporal overlap ..". Probably not easy to express in a few words but it does reflect importantly on the results, specially if the temporal variation of aerosols is considered. A graphical representation of the sampling timeline along with, say, total aerosol concentration may be useful.

| 334-335 | 'several prior works' and 'several publications' are unnecessary words |
| | |
| 379-382 | Bit unclear. Seems to indicate that tests were made during FIN-03 and the factor 90 increase found previously was confirmed. |
| | |
| 413-417 | The vagueness of this description leads to unease about the results. What is meant by 'non-ideal' behaviour? |
| | |
| 418 | "... was then applied ..."  When? |
| | |
| 422 | it is unclear what "linear interpolation" refers to. Between particle-free sampling periods? How often was that done? |
| | |
| 425 | " points that exceeded water saturation" seems to be an error |
| | |
| 439 | Sentence appears garbled and it is unclear what 'complementary' means here. |
| | |
| 442 | ".. distributed liquid particle suspensions .." - unclear |
| | |
| 466 | This is the first mention of the fact that off-line measurements were also off site. Please include that information at some more prominent point. |
| | |
| 470 | ".. detected by an optical microscope.." - needs better wording |
| | |
| 472 | The dependence on cooling rate is known and can be corrected for variations. If the other instruments used a different rate, the comparison can be improved by applying the known corrections. |
| | |
| 478 | Dilution has not been mentioned before. |
| | |
| 479-480 | More accurately, spectra are calculated for $f_{unfrozen}$ determined at 1ºC intervals. Not so? |
| | |
| 482 | Were blanks processed for NC State CS? |
| | |
| 508 | " ...special contribution ..." One wonders why this is considered special. |
| | |
| 508-540 | This material reads more part of an interpretation of results than a methods dscription. |
| | |
| 547 | Presumably, 'standard method' refers to measurement with this device, not in a more general sense. |
| | |
| 562 | Is the EAC used in FIN-03 the same as the PEAC7 described in the reference? More importantly, are the size-dependent collection efficiencies of this EAC known and were corrections made to account for it. This is specially relevant to INPs potentially of sub-micron sizes. |
| | |
| 586 | " ... attempted to operate ..." leads to questions |
| | |
| 588 | Presumably, "agreed upon" for reasons of surpassing detection limits and to get statistically reliable results. |
| | |
| 582 | Section 2.3 seems to have material that should be elsewhere. The sampling strategy should probably be clarified earlier on. See comment on line 316. The range of temperatures for inter-comparison is a matter of detection limits and, to some extent, a compromise with considerations of sampling times. |
| | |
| 725-740 | This brief reference to the slopes of the temperature spectra is, apparently, based on some visual fit of the curves in Fig. 5. More could be said about how this was done and much more should be said about variations in the slope with temperature and with time. The slopes at given temperatures show significant differences among instruments. |

| 726 | Is the MIT-SPIN data for a single measurement or average. If the latter, why no error range is indicated? This sentence should be in the preceding paragraph, not when the temperature trend of the data is discussed. |
|---|---|
| 733 | Unclear what is meant by ".. standard deviation of the measurement means ..". Is it mean values for the 1ºC interval? Once the data has been set at 1ºC bins, as stated earlier, they do not need to be called 'mean' values. It is also a bit unclear that the mean and standard deviation refer to averaging all values at a given temperature throughout the campaign. Assuming that, it would be helpful if here, or at some other place, the number of measurements represented by each point would be given. |
| 738-740 | Any suggestion for explaining the difference? |
| 743-745 | Can the difference be quantified and its significance assessed in relation to data scatter? |
| 747 | Wasn't the FRIDGE sampling inside the laboratory, while the other two on the roof? |
| 751 ... | Except for the online instrument, what is meant by average over 1ºC intervals? Why these 3-hour averages differ from the 4 hour sampling times? |
| 768 | Figure 6 is probably the most complete presentation of the intercomparison results, yet it is hard to extract much information from it. The vertical scales are very compressed, and too many lines are presented. Could fewer temperatures be included? The color code is overly subtle, perhaps lines for -20ºC, and -25ºC could be heavier to help orientation. |
| 772 | See question about 3-hour average for line 751. |
| 827 | " ... higher nucleation efficiency .." might better expressed as having higher likelihood of containing ice nucleation sites.. |
| 848-850 | Breakup in itself is not an obvious explanation for reduced ice nucleation activity, unless sites are assumed to be formed at component contact areas. Various other factors also come into play when particles are introduced into water. |
| 865-870 | This paragraph deserves a second look. First of all, error sources may be un-assessed but not ultimately unquantifiable. Second, cooling rate dependence is pretty well quantifiable for cold-stage devices. The effect of sample storage has been assessed as referenced on line 868. Instrument temperature drift is in contradiction with the stated accuracies. Filter and impactor sampling has substantial possible uncertainties for unknown INP sizes. Many other sample handling issues may be variable to some extent. |
| 880-884 | Do the p-values here refer to no discernible offset? |
| 899 | What does IS stand for? |
| 1186 | A bit of a stretch to indicate that the range of measurements extended to -5ºC. Comparisons were all for -15ºC and lower (Fig. 6) |
| 1201 | Closure may refer here to the aerosol/INP connection. INP/cloud-ice connection is another, so this distinction should be made. Regarding the aerosol/INP closure, there is a limit beyond the instrumentation complex here utilized in that INPs may constitute a subset of the aerosol different in composition and size than the predominant aerosol. |
| 1256 | The choice of wording ".. relative agreement .. " makes one wonder as to relative to what. Perhaps 'the degree of agreement' is less vague as it can be thought to refer to the degree established in the analyses. |

1181    The authors should consider including in the Summary some comments to weigh  the atmospheric relevance of the results with regard to the temperature range covered by the measurements and the uncertainties of the relation between INPs and ice development in clouds.

---

## Author Comment (AC1)

**Response to Referee comment by Gabor Vali on the "Fifth International Workshop .... " by DeMott and co-authors.**

*As a preface, we agree with Dr. Vali on many points in his summary. While the paper is long, we wished to report as comprehensively as possible on the procedures, results and insights on remaining challenges that can be gained from such exercises. Some excellent points are made in this review, and we have implemented changes related to these, as specified below. Our hope is also that with the data available to others in future efforts, even more might be learned than we were able to capture in this paper. All our responses below are italicized.*

This manuscript reports the results of a workshop held in 2015 at the Storm Peak laboratory in Colorado with a large collection of ice nucleation and aerosol instruments. The main goals were to assess the degree of agreement among ice nucleation instruments of various designs and to interpret the results with consideration of aerosol characteristics. These goals have been mostly achieved. The agreements found are significant and the authors also make clear where discrepancies appeared. The results are described in detail.

Since the process of ice nucleation is complex, inferences about how it takes place in the atmosphere or in the laboratory have been arrived at over many decades from the way the process is manifested in instruments that proceed from prior condition to the appearance of ice with controlled characteristics. There are inherent limitations to this approach as knowledge is hard to achieve of relevant aerosol properties and of the way conditions are altered on the way till observation of ice becomes possible. Yet, by gradual understanding of the impacts of instrument design approaches have been key to the gradual development of a degree of understanding of how to achieve usable measurements of heterogeneous ice nucleation by atmospheric aerosol particles and by some types generated from known substances.
The workshop described in this paper undertook to test how instruments which have been shown to yield agreements among themselves with controlled aerosols will perform in the relatively clean high-altitude environment sampling ambient aerosol. Similar efforts have been reported previously, but this paper reports on significantly more extensive measurements. No surprises emerged from taking the devices to high elevations and, perhaps, that can be taken as reassurance of their reliability.

The paper is long and demanding. Instruments and sampling setups are detailed, possibly to a greater extent than necessary since previous publications are available for all. On the other hand, the results are presented in a more compressed style. The aerosol data are used to interpret outliers in the intercomparisons but this is somewhat cursory. Going beyond the intercomparisons, testing agreements with assumed particle compositions and parameterizations (Section 3.5) could well have been a separate paper. That section and the aerosol data needed for it take up, by a rough estimate, about 1/3 of the paper. Similarly, the deposition mode data, from only one instrument, could be separated from the main subject.

*The paper is long, and it has been made longer in responding to the many important clarifications asked for in the review. Nevertheless, details are important and our effort to publish as much complete information now as possible is intended to provide a perspective to the large and still*

*growing community of experts in this field on the approaches taken in FIN and the status of the measurement field at that time. In fact, we believe that participation in FIN stimulated many of the subsequent efforts to perform longer-term comparisons of INP measurements. We also agree that the discussion of results was somewhat compressed, and therefore have added discussion and another figure in response to the specific and helpful comments in this review. We also consider, to be truthful, that one paper that attempts to show as much as possible is preferable to the chance that additional papers might not come to fruition within a time frame that makes them relevant. We see the added sections as framing topics that were wholly part of this workshop and might be expected to be part of any such intercomparisons in the future.*

There is little attention in the paper on the absolute values of INP concentrations observed. The mountain-top observatory could be expected to yield INP concentrations lower than those observed a low altitudes. This doesn't appear to be the case.

*This is a valid and interesting point, albeit one we do not wish to expand on presently. There have now been several studies at different elevation just in Colorado, USA, including the recent Department of Energy (DOE) SAIL campaign based over 20 months in Crested Butte, Colorado. Taken together with other DOE campaign collections and other international campaigns held over multi-month periods, this topic could be explored by others in the future using the FIN-03 and these other data sets.*

The main result of the intercomparisons of instruments is stated as an order of magnitude agreement for immersion freezing INPs. This is a good result from the perspective of instrument reliability compared to, say, what the situation was 20 years ago. On the other hand it is worth seeing it as a 5°C discrepancy for given INP concentration. For atmospheric cloud processes that is still a huge uncertainty. Thus, for INP measurements to be useful predictors of cloud evolution at given location and times there is much remains to be accomplished. This does not diminish the accomplishments demonstrated in the paper, but it may be worth reflecting on in the Abstract and in the Conclusions.

*We agree and have added a statement of such into the abstract and Summary section.*

On its own terms, the evaluation of prior parameterizations with the extensive aerosol data available from FIN-03 is a laudable idea. Combined with similar efforts a better appreciation is developed for the validity of the generally used predictive equations for atmospheric INP concentrations.

*We thank the reviewer for this comment. Combined with our motivations stated above, we wish to retain all sections on parameterizations and deposition nucleation studies.*

Essentially all data reported is for measurement temperatures of -15°C and lower. This limitation should probably made clear to the readers right up front. The reasons for this limitation also deserve to be stated. A further caveat applies to having all comparisons done in terms of the cumulative number concentration, with only brief attention to the slopes of the temperature spectra.

*Regarding the first point, we now write in the second paragraph of the abstract, "These comparisons were restricted to temperatures lower than –15 °C due to limits of detection related to sample volumes and very low INP concentrations."*

*In section 3.3, paragraph 1, we write, "Only two sets of instruments were able to explore the temperature regimes of –30 °C and colder due their design to permit operation there, or warmer than –15 °C due to detection limits (controlled by sample volume and drop size used for immersion freezing)."*

*We address the second caveat in response to specific comments.*

The degree of agreement among instruments is extensively analyzed with reference to Figs. 6 and 7. Another striking point that can be added to that analysis that while FRIDGE-CS and CSU-IS show significant changes with time between Sept 14 and 16, specially at the higher temperatures, these changes are not evident in the other data. Then, for the next two days, these two instrument show greater disagreement in Fig. 7.

*We address this in specific comments below, highlighting the switch in bias between the two systems earlier versus later in the study. This discussion appears in the revised discussion of these figures and a new added figure in the results section.*

In the title, placing importance on what was done, rather than on the vehicle for doing it would be useful, i.e. "Field Intercomparison of Ice Nucleation Measurements; the Fifth ....."

*We agree and have changed the title accordingly.*

Line numbers:

77-176    The second part of the Introduction is not as clear as it should be for readers not already familiar with the subject. More specific comments are given in the following.

120    The text of the preceding lines calls for a somewhat more detailed reference to history. Ice nucleation comparison workshops go back as far as 1967. The main reason for many repetitions is not a difference in goals but the development of new instruments and better characterization of the processes underlying the different instruments, i.e. better understanding of heterogeneous ice nucleation. Another important factor was growing recognition of having detailed aerosol characterization (lines 169-172) and better instruments for that information. All of these factors see advances, but important limitations also remain.

*Thanks, this is important. We add, "Ice nucleation workshops have a history to 1967, with repetitions occurring in 1970, 1976, and 2007 (DeMott et al., 2011). These exercises were repeated not due to a difference in goals but due to the development and improvement of new ice nucleation instrumentation and a focus on better characterization of heterogeneous ice nucleation processes. An additional*

*factor that has motivated formal and informal instrument intercomparisons is growing recognition of the importance of having coordinated detailed aerosol characterizations and better instruments to provide that information (Coluzza et al., 2017; DeMott et al., 2011, DeMott et. al, 2018; Knopf et al, 2021; Brasseur et al., 2022; Lacher et al., 2024)."*

121-150 Following on the previous comment, it would be helpful to readers to have a concise argument for why FIN-01 had to be followed by -02 and 03. Why was -03 deemed necessary, The contrast with Knopf et al (2021) and Lacher et al (2024) would be best detailed at the end of characterizing FIN-03 goals.

*We also agree on this comment. While we touch on most of this later in the introduction, we add, "While laboratory experiments can easily provide broad concentration ranges of particles of specific types for texting, measurements in the ambient atmosphere are the ultimate application of INP measuring systems, and the ambient atmosphere presents the most challenging measurement scenario due to sometimes very low INP concentrations and a host of INP source compositions."*

210-215 This may not be as simple as it appears here. Not all sizes of aerosol are of the same composition even at a given instant of time and the INPs are a very small subset of the total aerosol, so the manner in which aerosol composition is correlated with the INP measurements needs detailed description and substantial caveats. The subject should perhaps be addressed after the description of the instrumentation.

*We agree that this is a complex topic and effort. We mention that the exercise is informed by particle composition data, but we add the qualifier that we "attempt" to diagnostically determine the likely contributions of mineral and soil dust. We prefer not to create a separate special section in the methods regarding this topic, which is why we spend considerable time on it when we discuss parameterizations later in the results.*

235 " ... positive or negative mass spectra" - what does that mean?

*Corrected this statement to read "positive and negative ion mass spectra."*

236-241 This helps to clarify what was meant on lines 210-215, but only in part. Are the sample sizes of the two instruments the same?? If PALMS samples only a fraction of the particles sampled by LAS, how much uncertainty enters in the analyses?

*PALMS and LAS sampled the same aerosol stream in parallel and at the same time. Typically, the largest source of uncertainty in PALMS+LAS concentration products is statistical sampling uncertainty due to PALMS' limited sample set. Statistical uncertainties are described in detail in the Froyd et al. (2019) reference for aircraft studies using small sample periods (minutes). However, in FIN-03 long sampling periods (hours) reduced these statistical uncertainties to <<*

*the atmospheric variability and thereby introduce negligible additional uncertainty to the LAS concentrations.*

Section 2.1   The authors' approach to present how data from given instruments are interpreted along with the description of the instrument has merits. Caveats and limitations are stated up front. However, it separates those issues (some listed in the foregoing) from the actual interpretation of the results. In the end, I think it isn't working to the paper's advantage.

*We may not understand correctly, but we think the reviewer is suggesting holding off on how aerosol data will be used to assist interpretation of data from simple instrument description. The primary reason to keep the discussion as it is presently written is to keep the reader interested in why we even care about the particle data in the context of an INP intercomparison. We think this is an advantage.*

313   "All these instruments ..." refers to the types of devices or the specific units?

*All specific instruments used in FIN-03 were part of FIN-02, but a subset. We have clarified.*

326   Please clarify " ..substantial temporal overlap ..". Probably not easy to express in a few words but it does reflect importantly on the results, specially if the temporal variation of aerosols is considered. A graphical representation of the sampling timeline along with, say, total aerosol concentration may be useful.

*As we say later in the manuscript (section 2.3), intercomparison time periods were selected as 3-4 hour units (4 hours permitted for sampling, but then attribution to 3-hour units of the day was made for plotting data). If an INP sample was drawn within or overlapping that 3-hour period, it was taken as a consistent sampling. Figure 2c makes it clear that adjacent 3-hr periods rarely represented surface area changes of more than a factor of 2 in the size range > 0.5 um and was usually within 10-20%. Large differences across 3-hour periods were much less for surface area at smaller sizes. This seems relevant to the utility of the selection of time periods for comparison, so we now mention this.*

*In this section, we note that aerosol data assisted this definition, and we reference the later sections. We write, " This means that on a given day a sample was fully collected within the comparison time unit of 3 hours (informed by aerosol data, as discussed later) or overlapped the comparison period if the collection time was somewhat longer."*

*In Section 3.2.1 we write, "Finally, adjacent 3-hr periods rarely represented surface area changes of more than a factor of 2 in the size range > 0.5 um and was usually within 10-20%. Large differences across 3-hour periods were less frequent for surface area at smaller sizes. These factors confirm the validity of the selected intercomparison time periods."*

| 334-335 | 'several prior works' and 'several publications' are unnecessary words |
|---|---|
| | *Removed as suggested. We write, "The CSU-CFDC operating principles are described in prior works (Rogers, 1988; Rogers et al., 2001; Eidhammer et al., 2010). Application and considerations for interpreting data have been described by DeMott et al. (2018). "* |
| 379-382 | Bit unclear. Seems to indicate that tests were made during FIN-03 and the factor 90 increase found previously was confirmed. |
| | *The 90 was the factor found for FIN-03, specifically. This result can vary depending on the aerosol environment. Tobo et al. was in a somewhat lower elevation mountain forest environment in Colorado. Perhaps the aerosols overall are quite similar in these two settings. We do not want to belabor the point as discussed in previous publications but add some qualifying text to note that the factor can vary by location, as has been reported. "Concentration factors for INPs can vary depending on the ambient INPs present in a given environment. These were evaluated in the same manner as Tobo et al. (2013), leading to an average increase of INPs by 90 times during operation of the aerosol concentrator compared to ambient inlet periods during this study (not shown here because analysis repeats the efforts referenced above)."* |
| 413-417 | The vagueness of this description leads to unease about the results. What is meant by 'non-ideal' behaviour? |
| | *This is now clarified, as it represents a well-documented factor that can vary with instrument design and is presumed to relate to turbulence where particles enter the CFDC chamber, but always reflects particles ejected outside of the intended lamina by the failure to achieve smooth introduction without any mixing. This is discussed in detail in the noted publications, so here we attempt to keep it simple as: "Finally, a SPIN specific particle concentration correction factor of 1.4 is applied to account for non-ideal instrument behavior (e.g., out of lamina particles) resulting in underestimation of INPs as described by Garimella et al. (2017). As the field measurements from this study predate the laboratory experiments performed to determine SPIN uncertainties, the minimum reported correction factor was selected to remain conservative in reported measurements."* |
| 418 | "... was then applied ..." When? |
| | *A depolarization filter was "next" applied. It means that the depolarization data were used to isolate data that were specifically determined to be ice particles.* |
| 422 | it is unclear what "linear interpolation" refers to. Between particle-free sampling periods? How often was that done? |

*Yes, linear interpolation refers to interpolating the background INP concentration between filter periods for baselining data. A minimum of at least 4 filter periods of 5-10 minutes long were conducted for each experimental set-point temperature, over typical 1–3-hour periods. We write, " Frost ejected from the plates of the SPIN chamber beyond that removed by the low-pass filter was characterized using particle-free sampling periods when the sample flow was diverted through a HEPA filter by an automated three-way valve. Linear interpolation of filter period INP concentrations was used to approximate background frost concentrations throughout the measurement period (a minimum of 4, 5-min filter periods for each set-point temperature within a 2–3-hour period) and smoothed using a five-minute moving average. Sample data was background frost corrected by subtracting this smoothed background frost density from total number density in each 5-min sample period." Note that we have added a similar comment on the procedure used by the CSU-CFDC in that section of the paper as, "A typical daily cycle at each temperature point was to bookend 10-min ambient air sampling with 5-min filter periods."*

425      " points that exceeded water saturation" seems to be an error

*This is not an error. The SPIN has a shorter evaporation region compared to the CFDC and at the time of FIN-03 it was not operated up to a higher RH limit as done in more recent years. We write, "Lastly, data points for which RH exceeded 100% were excluded from analysis for the MIT-SPIN at the time of FIN-03, predating development of procedures to define ice at up to 102.8% in Lacher et al. (2024). Hence, any data reported herein for comparisons to immersion freezing data are at the 100% RH maximum."*

439      Sentence appears garbled and it is unclear what 'complementary' means here.

*Revised to "Offline methods have undergone many improvements in recent years and have been successfully used in a complementary manner for comparison to online methods in other recent intercomparisons (DeMott et al., 2017; DeMott et al., 2018; Hiranuma et al., 2015; Wex et al., 2015)."*

442      ".. distributed liquid particle suspensions .." - unclear

*We rewrite, "In FIN-03 particles were collected from the air using liquid impingers and filter samplers. Impinger liquid and water suspensions created from immersed filters were analyzed for immersion freezing of distributed droplet volumes using ..."*

466      This is the first mention of the fact that off-line measurements were also off site. Please include that information at some more prominent point.

*We add, "All of these measurements were made offsite after the return of impinger liquid and filters to the participant institutions, as done in most intercomparisons*

*of this type. The handling of samples is mentioned in regard to each instrument below." This is the appropriate place to mention it, as the subsections then state how the samples were handled, e.g., frozen and returned frozen."*

470        ".. detected by an optical microscope.." - needs better wording

*We write, "...freezing was detected at a temperature resolution of 0.17 °C (every 5 s) using images collected from an optical microscope."*

472        The dependence on cooling rate is known and can be corrected for variations. If the other Instruments used a different rate, the comparison can be improved by applying the known corrections.

*We now write, "Except for pure dust samples, the dependence of the population median freezing temperature on cooling temperature is less than 1°C per decade in cooling rate, including measurements of ambient INPs (Wright et al., 2013). A decade in cooling rate is much larger than the variations in cooling rate used by instruments in FIN-03 (-0.33 to 2 C min$^{-1}$). The expected shift in INP spectra due variability cooling rate is much less than the total uncertainty and thus cooling rate is not further considered here."*

478        Dilution has not been mentioned before.

*This seems the appropriate place to mention it, as most of the following instruments described will apply these same calculations. We are referring to serial dilutions needed to span a broad temperature range of measurements. We write, "where f accounts for any serial sample dilutions used at different measurement temperatures."*

479-480        More accurately, spectra are calculated for *f*$_{unfrozen}$ determined at 1°C intervals. Not so?

*We write, "The high temperature resolution freezing data were collected 3× per sample and f$_{unfrozen}$ was binned into 1 °C intervals for spectral calculations. "*

482        Were blanks processed for NC State CS?

*Yes. Thanks to attempting to confirm this question, we discovered that the data used by the coauthor who drafted an early version of this manuscript was not corrected for blanks that were included in archive files. We have therefore performed a reanalysis and describe that in the new manuscript. In this methods section, we write, "As for all INP samples in FIN-03, "blanks" were collected for each of the NCSU-CS sample types. The normal procedure for most blank filter assessments in the field is to momentarily expose a clean filter to flow in a collection unit. In the spirit of procedural testing that typifies workshops like FIN-03, a different method was trialed by the NCSU group., Ten filter "blanks" were specially collected on*

*days during FIN-03 by placing a 0.2 µm pore size filter as a backing filter to an 0.05 µm pore size filter in a secondary filter unit that was exposed to the same total ambient flow conditions as the primary INP filter unit. This 0.2 µm filter was processed the same as the primary INP filter (rinsed in 6 ml ultrapure water) and freezing results were presumed to provide a quite conservative estimate of filter background INPs. It was indeed found that the number of INPs per blank filter in these collections were much higher than for standard blank filter method used by the other groups. The results from the 10 blank filters were averaged across the processed temperature range, and an upper confidence limit of 1 ºC was defined. All INP concentration results for each ambient filter were rejected if in any given temperature bin they fell below this upper confidence bound. In sum, 20% of the original measurement points based on filter collections were removed from measurement intercomparisons by this blanking operation. Impinger blanks were collected via separation of some water from the pure water storage container each time the impinger unit was filled with pure water to begin an air sampling period. Thus, blanks were specific to each ambient sample. The same 1 ºC upper confidence bound that characterizes NCSU-CS measurements was applied in each case to identify sample temperature points where the liquid suspension INPs fell below the upper confidence limit of the impinger blanks. These were removed from intercomparisons."*

508   " ...special contribution ..." One wonders why this is considered special.

*Replaced "special" with "supplemental".*

508-540  This material reads more part of an interpretation of results than a methods description.

*We respectfully disagree. We are attempting to summarize a diagnostic method for analyzing the contributions of different materials to ice nucleation, a set of procedures that not all readers may be familiar with, but ones that are becoming more commonly practiced even though this was not the case in 2015 when FIN-03 was held. We emphasize that we are speaking of methods to be applied to selected FIN-03 samples.*

547   Presumably, 'standard method' refers to measurement with this device, not in a more general sense.

*We have removed the parentheses. "b) the diffusion chamber method (hereafter: FRIDGE-DC), that addresses the deposition nucleation and condensation freezing modes introduced in Schrod et al. (2016) and is the standard method for operating the FRIDGE device (e.g., DeMott et al, 2018)."*

562   Is the EAC used in FIN-03 the same as the PEAC7 described in the reference? More importantly, are the size-dependent collection efficiencies of this EAC

known and were corrections made to account for it. This is specially relevant to INPs potentially of sub-micron sizes.

*We do not see any size-dependent differences in the collection efficiency in any experiments (Schrod et al., 2016). Therefore, no size-dependent correction was applied. A general sampling efficiency correction factor was applied.*

586       " ... attempted to operate ..." leads to questions

*Replaced with "nominally operated". It was simply an agreement that to the extent possible, certain operational conditions would be produced using each device.*

588       Presumably, "agreed upon" for reasons of surpassing detection limits and to get statistically reliable results.

*That was one reason, which we now state. However, where processing can only occur later, one is somewhat blind to the needed sampling period. This was an educated guess, and the greatest concern was aerosol variability that has diurnal, other meteorological, and even unpredictable event (e.g., biomass burning and other transports) dependencies. Others that might make the discussion wander are the logistics of preparing and operating instruments daily. We choose to keep the discussion simple.*

582       Section 2.3 seems to have material that should be elsewhere. The sampling strategy should probably be clarified earlier on. See comment on line 316. The range of temperatures for inter-comparison is a matter of detection limits and, to some extent, a compromise with considerations of sampling times.

*This seems to be a philosophical choice in how and where material is introduced. To our view, making these statements just prior to the presentation of results is a proper transition. We have already discussed detection limits that might interfere in the methods section above.*

725-740   This brief reference to the slopes of the temperature spectra is, apparently, based on some visual fit of the curves in Fig. 5. More could be said about how this was done and much more should be said about variations in the slope with temperature and with time. The slopes at given temperatures show significant differences among instruments.

*Recall that for this figure these are average data for the project, meant to give the "lay of the land" so to speak and to allow general comparison to data from other sites for those wishing to reference the study. We do not wish to explore more in this section. We have added, "The dependence of $N_{INP}$ on temperature is nearly log-linear from –10 to -27 °C, excepting perhaps a steepening of slope from –20 to –25 °C and some lowering of slope below this temperature. "*

726   Is the MIT-SPIN data for a single measurement or average. If the latter, why no error range is indicated? This sentence should be in the preceding paragraph, not when the temperature trend of the data is discussed.

*The data point represents a single measurement period on September 23. There is no special place to discuss it, so it is mentioned for completeness at this point.*

733   Unclear what is meant by ".. standard deviation of the measurement means ..". Is it mean values for the 1℃ interval? Once the data has been set at 1℃ bins, as stated earlier, they do not need to be called 'mean' values. It is also a bit unclear that the mean and standard deviation refer to averaging all values at a given temperature throughout the campaign. Assuming that, it would be helpful if here, or at some other place, the number of measurements represented by each point would be given.

*We hope it is now clear that these data are indeed averaged for each instrument at each temperature for the entire study. We also now write at the start of this section: "A summary of the number concentrations of immersion freezing INPs ($N_{INP}$) over the course of the field campaign, for all measurements at one degree temperature intervals for each instrument, is shown in Figure 5."*

*This clarifies what is shown in this figure. As we had to modify the numbers for the reprocessing of the CS data, we now list the count of measurements in each mean as separate columns in the DOI-linked data for this figure. Each individual data point used in calculating the means and standard deviations is included in the linked data for original Figure 7 (now Figure 8).*

738-740  Any suggestion for explaining the difference?

*We do not have special insights as yet. It could be that more environments were folded into the DeMott et al. (2017) paper. We choose not to speculate.*

743-745  Can the difference be quantified and its significance assessed in relation to data scatter?

*This comes now in discussion of the revised Figure 7 (now Figure 8), a table of statistics about these ratios of measurements that is provided in new Table 2, and a new Figure 6 that presents some direct comparisons between instruments over the entire project that we now show before discussing the temporal fractions. New Figure 6 shows the direct relations between all other measurements of $N_{INP}$ and the CSU-IS measurements of NINP, and the slopes of the linear regressions are stated in the caption. While showing that the best relation occurs between the FRIDGE-CS and CSU-IS during the overall study, the temperature range of greatest discrepancies is also identified.*

[Figure]

**Figure 6**. (a) INP concentrations for all intercomparison measurement points of FIN-03 from the FRIDGE-CS, NCSU-CS (I), NCSU-CS (F) and CSU-CFDC compared to the INP concentrations from the CSU-IS measurements. Linear regressions with zero intercepts are color coded for each, having slopes of 0.69, 0.19, 0.13 and 0.16 for the FRIDGE-CS, NCSU-CS (I) and CSU-CFDC, respectively. (b) The same data are color coded for different temperature ranges in °C and the 1:1 relation is shown.

*Associated with this figure, we now write:*
*"To view the overall data in a more complete manner over the entire project, we explore direct comparisons of instrument versus instrument data as scatterplots and also on temporal bases. First, in Figure 6, we show a commonly used representation of large INP project data as INP concentrations for four instruments versus one other and segregate the data into broad 4-degree temperature ranges. The data used for normalization were from the CSU-IS, though we might have used any other. Linear regressions were plotted in Figure 6 to show the overall average differences between measurements that are already evident in Figure 5. Figure 6a thereby demonstrates the generally good correspondence between the NCSU-CS data of both types and the CSU-CFDC data that measure factors of 5 to 8 lower INP concentrations on average compared to the CSU-IS, as well as the closer correspondence of the FRIDGE-CS (31% lower) and CSU-IS data. Greatest variations in INP concentrations over the course of the project seems to be focused in the −20 to −25 °C temperature regime (Figure 6b), where variations reached nearly two orders of magnitude. This is not an uncommon observation, also seen in Lacher et al. (2024). Surprising, but not easily understood yet, is the fact that all measurement methods could at times measure equivalently to or more than the CSU-IS."*

*The temporal data in revised Figure 7 (now Figure 8) shows the key time periods where these discrepancies occur. This is elaborated on in the discussion of these results. Finally, the mean, standard deviation and confidence intervals of the ratio data (all temperature points included) in revised Figure 7 (now Figure 8) are listed in new Table 2 in a manner that mimics the order of panels in the figure. Count data for each time series is also listed in Table 2. This table makes it clear that average disagreement did not exceed a factor of 5.5.*

**Table 2.** *Count number, mean, standard deviation (St. dev.), and 95% normal confidence intervals for the $N_{INP}$ ratio data of Figure 8 in the main manuscript, including all temperature points. As for that figure, numerator instrument is on the upper horizontal scale and denominator instrument is listed on the vertical scale.*

| | | FRIDGE-CS | CSU-IS | NCSU-CS(I) | NCSU-CS(F) | CSU-CFDC |
|---|---|---|---|---|---|---|
| **FRIDGE-CS** | N | | 149 | 108 | 94 | 19 |
| | Mean | | 1.01 | 0.21 | 0.26 | 0.61 |
| | St. dev. | | 2.70 | 0.54 | 0.43 | 1.60 |
| | CI | | 0.43 | 0.10 | 0.08 | 0.72 |
| **CSU-IS** | N | 149 | | 128 | 117 | 29 |
| | Mean | 0.98 | | 0.18 | 0.20 | 0.26 |
| | St. dev. | 2.19 | | 0.52 | 2.34 | 0.92 |
| | CI | 0.35 | | 0.09 | 0.42 | 0.33 |
| **NCSU-CS(I)** | N | 108 | 128 | | 87 | 27 |
| | Mean | 4.70 | 5.47 | | 1.41 | 0.95 |
| | St. dev. | 9.30 | 10.62 | | 4.88 | 1.13 |
| | CI | 1.75 | 1.84 | | 1.02 | 0.42 |
| **NCSU-CS(F)** | N | 94 | 117 | 87 | | 19 |
| | Mean | 3.73 | 4.90 | 0.70 | | 1.40 |
| | St. dev. | 7.06 | 5.54 | 1.51 | | 3.04 |
| | CI | 1.42 | 1.00 | 0.31 | | 1.36 |
| **CSU-CFDC** | N | 19 | 29 | 27 | 19 | |
| | Mean | 1.61 | 3.81 | 1.04 | 0.71 | |
| | St. dev. | 2.63 | 8.97 | 1.94 | 3.29 | |
| | CI | 1.18 | 3.36 | 0.73 | 1.48 | |

747      Wasn't the FRIDGE sampling inside the laboratory, while the other two on the roof?

*The FRIDGE sample was from one of the SPL turbulent inlets, and the characteristics were discussed in Methods. As mentioned, these inlets have 50% cut sizes above 10 microns, and the arrangement of the filter collection for the FRIDGE was to allow nearly direct sampling onto the filters. This is not the same as the open-faced IS filter that might effectively capture even larger particles, and we cannot say how the orientation of the NCSU filters affected capture efficiency versus size. This is already discussed in the paper.*

751        ...Except for the online instrument, what is meant by average over 1℃ intervals? Why these 3-hour averages differ from the 4 hour sampling times?

*This comment was very helpful in clarifying and revising what we had done. In the original manuscript and discussion, means were only used for FRIDGE data reported at higher T resolution and for CFDC data for which multiple samples were collected at one temperature. We now use FRIDGE data for specific temperature bins, as for all other data. We have also completely revised this approach for the new version of Figure 6 (now Figure 7) and its discussion, as noted just above. Measurement errors as confidence intervals are now listed with the data used and made available for Figure 6. We do not show the uncertainties in Fig. 6 to make it most readable. Finally, we also clarify that the data were broken into 3-hour periods for this figure and Fig. 7, even though a 3–4-hour period was used to define an intercomparison unit. We needed to be a little flexible because of logistical issues in trying to perfectly align all sampling during the study.*

768        Figure 6 is probably the most complete presentation of the intercomparison results, yet it is hard to extract much information from it. The vertical scales are very compressed, and too many lines are presented. Could fewer temperatures be included? The color code is overly subtle, perhaps lines for -20℃, and -25℃ could be heavier to help orientation.

*Figure 6 (now Figure 7) has been revised as suggested. It would take much more space to decompress scales. We think these are clear, and discussion is provided.*

[Figure]

772        See question about 3-hour average for line 751.

           *The plotting intervals are 3 hours, which we now clarify. If the sampling occurred*
           *fully over 4 hours, the period it most fit into is used as the x-scale.*

827        " ... higher nucleation efficiency .." might better expressed as having higher
           likelihood of containing ice nucleation sites..

           *Good point. We have implemented it. We write, "Larger particles do tend to have*
           *higher likelihood of containing ice nucleation sites,..."*

848-850    Breakup in itself is not an obvious explanation for reduced ice nucleation activity,
           unless sites are assumed to be formed at component contact areas. Various other
           factors also come into play when particles are introduced into water.

*We agree that there are many factors that can come into play when particles are introduced into bulk water. We were referring to disaggregation as a mechanism for increasing INP activity for largest particles for which multiple sites might have otherwise been masked, including the most active ones. We add, "For example, if very large aggregates that are preferentially collected by one substrate versus another, disaggregation in water could lead to a high bias in ice nucleation sites effective at lower temperatures."*

865-870    This paragraph deserves a second look. First of all, error sources may be un-assessed but not ultimately unquantifiable. Second, cooling rate dependence is pretty well quantifiable for cold-stage devices. The effect of sample storage has been assessed as referenced on line 868. Instrument temperature drift is in contradiction with the stated accuracies. Filter and impactor sampling has substantial possible uncertainties for unknown INP sizes. Many other sample handling issues may be variable to some extent.

*Thanks for this comment. We have discussed size-dependent sampling issues already in the preceding paragraph. Here we change what is written to, "Besides size-dependent sampling biases, the fact that measurements of immersion freezing INP concentrations from ambient air can be uncertain by up to one order of magnitude may result from unquantifiable random or non-random factors, or more likely from quantifiable factors that were not fully controlled in this field study nor easily controlled across investigating teams in general. Examples of known issues that were only documented after FIN-03 relate to inconsistency in sample materials or sample handling and storage (e.g., Barry et al., 2021b; Beall et al., 2021)."*

880-884    Do the p-values here refer to no discernible offset?

*The p values indicate that the observed correlations are not due to chance, which is expected since the instruments all appear to successfully measure INP concentration, at least to some degree.*

899    What does IS stand for?

*Corrected to "CSU-IS".*

1186    A bit of a stretch to indicate that the range of measurements extended to -5ºC. Comparisons were all for -15ºC and lower (Fig. 6)

*Agreed. We have revised the sentence to factually represent that archived measurements were made over this temperature range by one or more instruments and that two or more instrument comparisons were analyzed between –15 to –30C. We write, "The immersion freezing INP concentrations measured in FIN-03 by one or more instruments spanned a dynamic range of over five orders of magnitude ($10^-$*

*³ to ≈102 L⁻¹) over the temperature range –34 °C to –7 °C. Intercomparisons of two or more measurements were made from –30 to –15°C."*

1201    Closure may refer here to the aerosol/INP connection. INP/cloud-ice connection is another, so this distinction should be made. Regarding the aerosol/INP closure, there is a limit beyond the instrumentation complex here utilized in that INPs may constitute a subset of the aerosol different in composition and size than the predominant aerosol.

*This is very well stated and we now use these phrases to write, "Although FIN-03 was not conducted as an aerosol/INP closure study per se...", and at the end of the paragraph, "Nevertheless, there is a limit beyond the instrumentation complex here utilized in that INPs may always constitute a subset of the aerosol different in composition and size than the predominant aerosol. Knowledge advance may require improvement in methods that link INP and compositional measurements on single particles to specifically isolate these factors. Hence, a great amount of work is still needed..."*

1256    The choice of wording ".. relative agreement .. " makes one wonder as to relative to what. Perhaps 'the degree of agreement' is less vague as it can be thought to refer to the degree established in the analyses.

*We add words now to be more explicit. The paragraph now begins, "In summary, the agreements amongst instruments during FIN-03, within factors ranging from nearly 1 to up to 5.5 times on average between individual measurements and rarely exceeding one order of magnitude in short time periods, match those found in the FIN-02 laboratory studies. These represented state-of-the-art for measurements at the time of FIN-03 and taken together with further improvements since this time as reflected in recent studies (Knopf et al., 2021; Brasseur et al., 2022; Lacher et al., 2024) demonstrate steady improvement in the community's collective ability to detect and quantify atmospheric ice nucleation."*

1181    The authors should consider including in the Summary some comments to weigh the atmospheric relevance of the results with regard to the temperature range covered by the measurements and the uncertainties of the relation between INPs and ice development in clouds

*While we prefer to focus on the aerosol/INP aspects of closure, we add a few sentences now at the end of the Summary. "There is a clear need in the future to extend measurement comparisons to the atmospherically-relevant and critically important temperature range higher than –15 °C. The low atmospheric number concentrations of INPs existing at times at these temperatures is a significant challenge for such, reflected in this study by the inability to measure INP concentrations above detection limits at the SPL site even for 3-to-4-hour filter collections at temperatures higher than –7 °C. Longer sample times and higher*

*volume collections can improve this situation, but introduce other technical challenges and do not appear possible for online instruments.*

*We also herein do not address the relevance of INP measurements overall for understanding ice formation in clouds, where secondary processes may come into play. This is an additional topic for critical investigation, given a degree of confidence now established in measuring INPs. However, the fact that 5-factor to order of magnitude correspondence between measurements equate to 3.5 to 5 °C temperature uncertainties in assessment of INPs is something that also deserves scrutiny from the cloud modeling community concerning if this is satisfactory, and if not, what level of correspondence should the INP research community be seeking."*

---

## Author Comment (AC2)

**RC1 review of "The Fifth International Workshop on Ice Nucleation Phase 3 (FIN-03): Field Intercomparison of Ice Nucleation Measurements," by P. J. DeMott et al. 2024**

**RC1**: This manuscript is well written. The authors conducted a fair intercomparison of online and offline INP-measuring instruments in the field. Despite the challenging environment at SPL, invaluable outcomes and lessons are reported in a neutral and unbiased manner. Furthermore, the authors include a list of limitations (e.g., deviation in sampling particle sizes etc.) and things to be further explored in this manuscript for more understanding of aerosol-cloud interactions (e.g., a need for online/direct deposition ice nucleation measurements), which are important messages to the INP research community. This reviewer agrees that more research is necessary to predict and explain the temporal variation of biological INPs (perhaps in a predominantly biogenic environment). While the authors found a predominant contribution of mineral and/or other inorganic particles to INP abundance in this study, they also note the need for in situ mixed INP detection and characterizations, especially for Soil & BBA INPs, which is important. The study topic is relevant to the journal scope of AMT. This reviewer supports the publication of this paper in AMT after the authors address several questions below.
*We thank the reviewer for this overall positive assessment of the paper.*

Questions

[1] Figure 7: This reviewer wonders if using 3-hr INP median or log-average changes any conclusions of this intercomparison study. The ratio in Fig. 7 is computed by using time averages, which is reasonable. But, since the reported $N_{INP}$ spans a log range at a majority of freezing temperatures examined in this study, the average can be biased by high $N_{INP}$ values at the given temperature, such as the ones from FRIDGE-CS and CSU-IS. Perhaps, using the median may overall result in better agreement for NC State(F), NC State(I), and CSU-CFDC? The same average vs. median argument applies Figs. 8 & 9.

*We thank the reviewer for this comment and these suggestions. They brought to the fore that we have not properly described the somewhat varied nature of data in Figure 7, which we do now in the revised manuscript. Primarily, it was erroneous to say that all data were averages. In fact, most data are single temperature measurements during cooling rate scans conducted on suspensions from single few-hour filters in the case of standard immersion freezing devices, and only in the case of the CFDC data are multiple point measurements averaged. Hence, converting to median values is not possible. Former Figure 7 (now 8) has been revised and other figures have been added or revised based on comments to Dr. Gabor Vali's review.*

*Figures 8 and 9 represent instrument averages and so are amenable to a median analysis. However, the number of overlapping periods of more than 2-3 instruments is minor and in analysis of medians for these cases, they differed from means by less than 20%. Hence, we retain the analysis as shown in the present figures.*

[2] Figure 5: This reviewer wonders why NC State CS(F) shows a lower detection of $N_{INP}$ (~6 x $10^{-3}$ $L^{-1}$) than NC State CS(I) (~$10^{-1}$ $L^{-1}$). The sampling air flow rate seems similar for these two methods as described in Sect. 2.2.2. The sampling interval was shorter for impinger sampling?

Or it may be due to the difference in collected particle sizes (L836-839; L846-848; L855-858)? This reviewer is aware of a general statement in L865-870.

*We thank the reviewer for helping us to clarify what is already apparent in the noted figure, that is, the detection limit is different for the two NC State measurement methods, and this is a consequence of differences in the liquid and air volumes used. To make this clear before results are shown, we write at the end of the Section 2.2.2 subsection on the NC State methods: "Note that due to the greater $V_{liquid}$ used in the impinger for the stated air collection volumes means that the filter samples were more concentrated by a factor ≈11. Thus, the filter technique is more sensitive and has a lower limit of detection (LOD). The precise ratio for any given experiment depends on the exact sampling times of filter and impinger, and the exact number of droplets for the filter, impinger sampling, averaging across repeats, and binning into 1-degree intervals. For this reason, the ratio of LOD for the averaged samples may differ slightly from this estimate"*

[3] P31L649-655: Low AE (<1) seen in 9/14-16 in Fig. 3b may be due to the predominance of large dust seen in Fig. 4? The authors also report that the submicron particles dominated during the study period (L-637-638). The effective aerosol scattering efficiency from SPL during this intercomparison campaign can be similar to what is reported in Testa et al. (2021)?

Refs.

Russell. P. B. et al., ACP, 10, 1155-1169, https://doi.org/10.5194/acp-10-1155-2010, 2010.

Testa, B. et al. JGR-A, 126, e2021JD035186. https://doi.org/10.1029/2021JD035186, 2021.

*We believe that the reviewer meant Fig. 2c instead of 3b, and Fig. 3 for Fig. 4. While a general preponderance of dust was possible during the period noted, dust seen in Fig. 4 on the 15th is from a very short period of PALMS operation and likely represents an anomaly due to generation of road dust near the site at that time, as discussed already in Section 3.2.2. There was scarce data from mass spectrometry during that period and so we choose not to emphasize it. While scattering efficiency might be like Testa et al. (2021), we do not wish to emphasize derivation of surface area from nephelometer data in this publication, as was done in Testa et al.*

[4] P49L971-973: Can the authors clarify this part?

*Yes, the intended meaning was that it would be unrealistic to believe that all particles in the size regime larger than 500 nm are soil-sourced only. The intention of the discussion is to say, what if they somehow were? We have revised this to read, "In this case, a somewhat unrealistic maximum assumption on soil dust numbers and surface area that considers all particles and compositions in this size range as emanating from dust, Niemand 2012 estimates a dust source for 50% and DeMott 2015 estimates 25% of observed INPs on average."*

Comments

P37L749-750: This is good. Comparability of impinger and filter-based methods shown in this work implies that ambient particles collected on filters are well-scrubbed in liquid suspension for

freezing tests on NC State CS, resulting in comparable $N_{INP}$ to that from directly suspended impinger samples, for this field study at least.

*We add to reflect this point by expanding the sentence to say, "…, suggesting that particle removal from filters can be highly effective for immersion freezing measurements of ambient particles."*

P44L885-887: This recaps that the link between aerosol chemical composition and INP is not straightforward and underscores the importance of ice residual composition data.

*Yes, although ice residual composition data is difficult to obtain due to the low INP concentrations one attempts to assess via that method, and the low efficiency of doing this by mass spectrometry, as discussed in limited publications on this topic since 2004. There can also be pitfalls for identifying particle types via SEM and TEM. It is hard work, though we agree that it must continue. We feel that this paper is not the venue for emphasizing this point though.*

P64L1249-1255: This reviewer agrees. The ultimate future INP instrument intercomparison may be performed on the aircraft platform in cirrus and/or pyrocumulonimbus cloud regimes with collocated aerosol instruments suggested by Burrows et al. onboard then.

*We appreciate the point the reviewer is making also but will only comment in this response. Aircraft campaigns are notorious for not providing enough signal to noise in comparison to ground based efforts. This is true for both INPs (typically lower, except perhaps in a pyrocu, though few pilots will fly into them) and with compositional measurements. Hence, while we agree that such intercomparisons would be ideal, they may be a work in progress over many years.*

---

## Author Comment (AC3)

**RC2 review of "The Fifth International Workshop on Ice Nucleation Phase 3 (FIN-03): Field Intercomparison of Ice Nucleation Measurements," by P. J. DeMott et al. 2024**

It was a pleasure to read "The Fifth International Workshop on Ice Nucleation Phase 3 (FIN-03): Field Intercomparison of Ice Nucleation Measurements," by P. J. DeMott et al. 2024. The manuscript presents data and summarizes results from the third of a series of instrument measurement comparisons that were aimed at achieving better collective community understanding when it comes to measuring ice nucleating particles (INPs). The focus of the third effort was utilizing a subset of instruments used in the first two phases, to focus on making field measurements of ambient air. While many of the same instruments were used in FIN-021, that study's focus was on laboratory-based measurements. The field-based FIN-03 measurements add complexity due to the heterogeneous nature of ambient aerosol and the low naturally occurring concentrations of INPs. The FIN campaigns are well known and brought much of the community together, but given the summarized measurements were conducted in 2015 it has been a bit of a wait to see the outcomes of the 3rd phase published. The results summarized in the submission remain well worth reporting and valuable to the community at large. However, in its current form the manuscript lacks polish and consistency, which diminish its readability. Moreover, given the large gap between when the measurements were made and their submission in this form, the submitted work seems to, in places, lack connections to work published in the intervening years. This is not a blanket statement, as some more contemporary results (references) are used, but there is a lack of consistency that again points to a need for a systematic review of the manuscript.

I suggest that the author's take into consideration the contextualized comments I include below and resubmit an updated manuscript.

*We thank the reviewer for the constructive comments regarding manuscript readability and have made what we consider strong efforts to revise accordingly and update other publications referenced. This was needed because the paper had a few evolutions, ultimately slowed by the pandemic, before it came to fruition. Many of these changes are discussed below in response to itemized comments. As here, all our replies are italicized.*

**Itemized Scientific and Editorial Comments:**

Suggestions are given by line number taken from the downloaded pre-print PDF document:
• (Abstract, 48) strike "a subset of"
• (Abstract, 50) suggest a rephrasing to: Composition of the total aerosol was characterized using ...
• (Abstract, 55) suggest a rephrasing to: Mineral dust containing particles were ...
• (Abstract, 56) should probably be diameters

*All above modified as suggested.*

• (Abstract, 58) Here and throughout the text (lines 93, 186, 187 etc.) the symbol ~ is used where

≈ would be a better choice. Although not always strictly followed, generally the former, 'similar to', means the same order of magnitude. Most often the author's intention it seems is the later, which is 'approximately'.

*We have modified to use "≈" throughout the manuscript, where appropriate.*

• (Abstract, 67) Should this say: ....order of magnitude or more, more efficient
• (Abstract, 76) strike "at most times"

*All above modified as suggested.*

• (79) suggest a rephrasing to: Aerosol particles that ...

*We have changed this to "Particles that…"*

• (140-143) A reference (and probably in other places) to Brasseur et al.[2], seems like it should be included. Much like the campaign summarized by the Lacher et al. 3 paper, the Brasseur et al.[2] Measurement Report describes intercomparison and measurements by instruments for counting INPs, including a subset of the instruments used in FIN-03.

*We were neglectful in omitting Brasseur et al. (2022). This reference has been inserted in all relevant places. A rigorous comparison between online and offline instruments was not discussed in Brasseur et al. (2022), which is why we have emphasized reference to Lacher et al. (2024) in this context.*

• (171) suggest to replace "aerosol" with particle.
• (228) A Thomson et al. (2000) paper is referred to but is not found in the listed references.
• (323-324) suggest a rephrasing to: ...all ice nucleation instruments utilized in FIN-03 is provided in Table 1. Detailed operating principles, locations of samplers ....

*All corrected as noted.*

• (362) "Frost corrections are defined through..." The wording is strange here. Suggest replacing through with, using or utilizing, or somehow rephrasing.

*We write, ""Frost corrections are defined via use of time intervals sampling..."*

• (§2.2) A general comment on the presentation of instruments and instrumental setups in Åò2.2, both §2.2.1 and §2.2.2. I would suggest the author's do a careful re-reading of the instrument sections and that they re-write the sections such that equivalent information is presented for each instrument. During reading the lack of consistency became most apparent in §2.2.2, but this forced me to return to the earlier text and notice a general lack of consistency. When presenting different instruments different details are presented, and it is not evident why or what, if any, importance these differences indicate. For example, for the NC State - CS the temperatures of sample storage are included, but not in the case of the CSU-IS. My intent is not that the author's should include all the information or all the instruments. Rather the author's should decide on

what are the important and relevant parameters and make sure to present them uniformly. The reader should have the same information for all apparatuses. Alternatively, they must tell the reader why some things are important for some instruments but not for others. The current presentation is both a-systematic and simultaneously uninformative as to why/how the information that is presented is chosen.

*We thank the reviewer for this comment. Similar comments were made by other reviewers. We reflected strongly on these and have made many changes in how information is reported and organized in each Methods subsection. We note that on rereading the sections carefully, they do proceed with the same outline of information. Further, this information needs to account for differences in online and offline methods. The online instrument sections begin with operating principles and procedures, sampling and inlets, discuss uncertainties in calculated INP concentrations and related corrections for false counting of non-INP, and finish with any special studies that will be reported. The offline sections describe the configurations for sampling, computation of INP concentrations and confidence intervals, and any special applications reported. We have attempted to assure now that all the same factors are discussed for all instruments, adding any missing information, modestly reorganizing and removing extraneous information.*

• (370) "(by 3 time)" by a factor of 3?

*Changed as suggested.*

• (378 and again at 407) The o.d., which I take to mean 'outer diameter' is an irrelevant dimension here, as it gives no information concerning inner diameter.

*We have replaced this statement with 0.19" inner diameter and have done this throughout the manuscript.*

• (411-412) The "low-pass filter" sentence seemingly comes from nowhere? Why are these counts removed? I do not follow the logic at the beginning of this paragraph.

*This has been changed to read more descriptively as, "A low-pass filter was applied next to remove all 1 Hz data that exceeded a total of three counts s$^{-1}$, as recommended by Richardson et al. (2007) to reduce obvious frost noise that equates to INP concentrations larger than about 200 L$^{-1}$ (>2 standard deviations above mean values discussed later) for the SPIN volume sampling rate." The reference to Richardson et al. (2007) refers to their discussion (using "IN" for "INPs")- "During IN measurements under conditions supporting only heterogeneous nucleation, 1 Hz OPC counts are dominated by counts of 0 and 1. Frost events are acute and generate 1 Hz OPC counts exceeding approximately 3 (roughly equivalent to 120 IN per liter). To prevent overestimation of IN concentrations, the 1 min data set was further corrected by setting these high 1 Hz events to 0 and re-averaging the periods in which these events occurred." This filtering of data is done on all data before lower levels of frost are corrected by comparing time periods on and off the HEPA filter. Because the CSU CFDC used an aerosol concentrator to improve statistical sampling at times and generally experienced only lower frosting issues during FIN-03, the low pass filter method was not applied in processing those data.*

• (447) suggest: ...has been previously...
• (462) suggest: ...except that Teflon tape replaced stopcock grease sealing the impinger...
• (491) suggest: ...2 m distant.
• (501-502) suggest: ...suspension estimated using Eq. 1. (Vali has been cited when presenting theequation.)
• (549-550) suggest: ....specific analytical procedures differ.
• (560) suggest: ...is derived using Eqs. 1 and 2.

*All above corrected as suggested.*

• (563-566) This is a confusing description of the EAC's working architecture and may not even be necessary given the instrument papers available. I would suggest re-writing if it is to be included. For example, what does "12 kV against a grounded..." mean? I think that it must be that the 12 gold wires have an applied 12 kV electric potential.

*We have revised and shortened this section to read, "Within the EAC aerosol particles are electrostatically precipitated onto silicon wafers, which are used as sample substrates. After sampling is completed, the analysis at select pairs of temperature and relative humidity set points follows in a separate step."*

• (576) suggest: At SPL samples were taken with the EAC for ...

*Changed as suggested.*

• (Figure 2, plus caption) The font in panels (a) and (b) is extremely small and hard to read, make uniform with (c) which is much better. (c) is entirely missing from the caption, although I take it that it is described beginning with, "Timeline...."

*We hope that the revised figure and caption are greatly improved. See below.*

[Figure]

***Figure 2.*** *Time series of dry particle number concentration distribution (ambient conditions, not STP) measured by the laser aerosol spectrometer (LAS) in a), shown as three-hour means at ambient pressure. Time series of particle surface area distribution is in b). c) Timeline of nephelometer scattering (1-hr data) in the red channel for < 1 mm and 1 - 10 mm size ranges, 3-hr LAS number concentration > 0.5 mm, 3-hr LAS surface area at sizes below and above 0.5 mm, and Angström exponent (dashed, right axis).*

• (666) extra (
• (759-760) suggest: ...during a period of warming

*Above two points fixed as suggested.*

• (766) I find it strange that figure S3 is referred to before figure 7 appears in the text, as S3 seems to be a distillation of Figure 7. It seems the authors should carefully consider their choices as it regards these results and figures. Moreover, both aforementioned figures strictly speaking present the 1:1 comparison incorrectly. In Figure S3 a blue FRIDGE-CS bar should show 100% agreement in the FRIDGE-CS column, and so on for all instruments compared with themselves. Currently they are missing, and the caption does not mention that the 1:1 comparison is ignored because it is 100%. The same is true of the diagonal of the figure matrix in Figure 7. In fact, S3 is a bit of a strange presentation, and perhaps some sort of matrix like presentation with a heat map (for example) would be better, but "agreement within 1 order of magnitude" is a bit awkward in general. Figure 7 is more informative but quite busy, and this might be the better figure for the supplement. Albeit I would include 1:1 or 100% agreement along the diagonal for rigor. In general, I would suggest this information could be better presented and communicated and would suggest the author's re-imagine these figures.

*Fig. 7 (now Fig. 8, with reference to the response to the review of G. Vali) is referred to prior to Fig. S3 in the manuscript, but they are mentioned together since their discussion goes hand in hand.*

*As for the figure selection for the main manuscript and supplement, we would rather have the complete data set in the main paper (Fig. 7, now 8). We do now include the data from Fig. S3 as a table (new Table 3) in the main manuscript.*

*For the self-comparison of instruments, we choose to leave this out, as it appears now. This was a conscious decision to make the figure a little less busy. We do add a mention in the caption now that these comparisons are omitted.*

*We also find the quick "one order of magnitude" information easy to grasp, and easy to compare to other intercomparisons from previous endeavors, so we would argue to keep it. However, we may note that we have added quantitative analyses of the results in new Figure 8 as Table 2. This demonstrates average agreements from 1 to 5.5 for all matched measurement pairs of instruments over the entire study (see response to G. Vali). The question raised by this reviewer (point: page 40) and by G. Vali about whether one order of magnitude is really "good" agreement is of course open to argument. We add some discussion of this point in the Summary section in response to the review by G. Vali. We include the temperature uncertainty represented by the results as well.*

*We did attempt several ways to reimagine these figures as heat maps, but these would turn out to be quite patchwork, which is the reason that a figure folding in all overlapping data (all temperatures in common) in one (Fig. S3) was used in the first place.*

• (Figure 6) Are all temperatures necessary? For the main text perhaps a distilled figure focusing on a few temperatures, or results for temperature averages could be a better alternative and more effective at communicating the point. Also, some lines connect points across missing data points at the warmest temperatures, where I assume no activation was observed. I would suggest that

these points are better left unconnected, for clarity of the figure, but also because it is likely the value is actually 0 INP or below the resolution. For the CSU-CFDC panel (e) such data points are not connected, which is a much better approach than in some of the upper panels.

In the figure caption, use the acronyms as they were introduced in the text. Strike "from Goethe University Frankfurt". This kind of information appears in the text and does not need to be reproduced in a caption.

*We agree that Figure 6 could be improved by limiting the amount of data shown, so have redone the figure with half as many temperature points and no lines connecting data points (these were only intended to show trends for the eye but fail for reasons mentioned and others). Because CFDC data was primarily collected at select temperatures and with only few data points made at intermediate temperatures, the primary temperatures shown for these data are -30, -25, -20, and -15 ºC) in this panel, with slight but proximal colors used to compare to the data shown only at odd temperatures for other instruments. We prefer not to take temperature averages. The new figure appears as below:*

[Figure]

• (page 40) A general question when using the comparison that 1 order of magnitude is a good agreement. It would be worthwhile to present a simple calculation to quantify how many particles (i.e., activated droplets or volumes in immersion freezing) result in an order of magnitude change. At some temperatures and concentrations, the counting statistics may be rather poor, it would be interesting to address this given the already dilute nature of the ambient sampling.

*This is not a straightforward exercise, as might be imagined, because different instruments have different number of droplets, droplet volumes, volumes of washing water, and air sample volumes, and therefore the number of freezing events resulting in a one order of magnitude difference would be different for each instrument and be different for different temperatures (number of unfrozen droplets). It is the case though that counting statistics only become an issue at the highest temperatures when limits of detection are approached. We discussed this in reply to the RC1 review in relation to understanding apparent differences between the onset temperature of freezing detectable by the NCSU impinger and filter methods. Otherwise, confidence intervals are described for each instrumental method, and these are never as large as one order of magnitude.*

• (Table 2) I see the CF constant discussed in the text, but no discussion of the other constants in the Table. Are these taken from previously published literature? Or used as fitting parameters here in some minimization scheme? Please illuminate.

*It was implicit that these constants listed were described/derived in the publications listed in the table.*

• (989) "trends better" what is meant here? correlates?

*This has been revised to, "The structure of the timeline of predicted $N_{INP}$ resembles that of the observed $N_{INP}$ only below -20 ºC..."*

• (1200) again the Brasseur et al.[2], seems like it should be included.

*Added as suggested.*

• (1240-1241) Rephrase, "factor of a few increases" Something here is awkward and unclear. Deposition freezing increases with increasing RH?

*We rephrase to: "The deposition INP concentration obtained by FRIDGE-DC increases from 95% RH to 99% RH on average by a factor of 3.3."*

• (1246) Again awkward phrasing, "...achieved statistical significance from the CSU-CFDC data."

*Revised to, "For the online instruments, only limited periods of deposition INP measurements with the CSU-CFDC achieved statistical significance."*

• (References) In addition to the previously mentioned Thomson text, the Burrows, 2022 paper that is referred to is not found in the Reference list.

*We thank the reviewer for these notes. All have been corrected.*